# *Toxoplasma gondii* infection drives conversion of NK cells into ILC1-like cells

Eugene Park[1], Swapneel Patel[1], Qiuling Wang[2], Prabhakar Andhey[3], Konstantin Zaitsev[3,4], Sophia Porter[3], Maxwell Hershey[1], Michael Bern[1], Beatrice Plougastel-Douglas[1], Patrick Collins[3], Marco Colonna[3], Kenneth M Murphy[3], Eugene Oltz[3,5], Maxim Artyomov[3], L David Sibley[2], Wayne M Yokoyama[1]*

[1]Division of Rheumatology, Department of Medicine, Washington University School of Medicine, St. Louis, United States; [2]Department of Molecular Microbiology, Washington University School of Medicine, St. Louis, United States; [3]Department of Pathology and Immunology, Washington University School of Medicine, St. Louis, United States; [4]Computer Technologies Department, ITMO University, Saint Petersburg, Russia; [5]Department of Microbial Infection and Immunity, Ohio State University Wexner School of Medicine, Columbus, United States

**Abstract** Innate lymphoid cells (ILCs) were originally classified based on their cytokine profiles, placing natural killer (NK) cells and ILC1s together, but recent studies support their separation into different lineages at steady-state. However, tumors may induce NK cell conversion into ILC1-like cells that are limited to the tumor microenvironment and whether this conversion occurs beyond this environment remains unknown. Here, we describe *Toxoplasma gondii* infection converts NK cells into ILC1-like cells that are distinct from both steady-state NK cells and ILC1s in uninfected mice. These cells were Eomes-dependent, indicating that NK cells can give rise to Eomes⁻ Tbet-dependent ILC1-like cells that circulate widely and persist independent of ongoing infection. Moreover, these changes appear permanent, as supported by epigenetic analyses. Thus, these studies markedly expand current concepts of NK cells, ILCs, and their potential conversion.
DOI: https://doi.org/10.7554/eLife.47605.001

*For correspondence:
yokoyama@wustl.edu

**Competing interests:** The authors declare that no competing interests exist.

## Introduction

Innate lymphoid cells (ILCs) comprise diverse populations. Like T and B cells, they are derived from common lymphoid progenitors (CLPs) but do not undergo antigen receptor recombination (*Spits and Cupedo, 2012*). ILCs are tissue-resident within mucosal sites where they participate in tissue homeostasis and survey for pathogens (*Sojka et al., 2014a*; *Gasteiger et al., 2015*), although recent studies showed that inflammation can induce ILC2 mobilization (*Huang et al., 2018*). ILC classification was initially based on their cytokine production, with type 1 ILCs (ILC1s) producing IFNγ, ILC2s producing IL-5 and IL-13, and ILC3s producing IL-17 and IL-22. ILC classification was subsequently corroborated by distinctive markers and developmental requirements.

Prior to the discovery of ILCs, NK cells were the only known innate CD3⁻ lymphocytes. Their classification within the larger scheme of ILCs has been challenging, in particular due to their strong resemblance to ILC1s while also maintaining important distinctions. For example, mouse ILC1s and NK cells both express NK1.1 and NKp46 and produce IFNγ, and as such were initially classified together as Group 1 ILCs (*Spits et al., 2013*). However, grouping NK cells with ILC1s was confounded by several discordant features. Whereas ILC1s are restricted to organs, and CD49a expression is a faithful marker of tissue-resident populations under steady-state conditions (*Peng et al., 2013*; *Sojka et al., 2014a*), most NK cells freely circulate. These cells also differ in their

developmental requirements, with NK cells requiring Eomes for development, while Eomes is not expressed in ILC1s and is dispensable for development (*Daussy et al., 2014*). Moreover, ILC1s are completely Tbet-dependent (*Daussy et al., 2014*; *Sojka et al., 2014a*), whereas NK cells require Tbet only for maturation (*Townsend et al., 2004*; *Gordon et al., 2012*). Consistent with their differential requirements for T-box transcription factors, NK cells and ILC1s arise at different branch points downstream of the CLP. While NK cells originate from NK cell precursor cells (*Rosmaraki et al., 2001*), other ILCs further differentiate along a pathway that includes the common helper innate lymphoid progenitor (CHILP) and ILC precursor (ILCP), which do not have NK cell potential (*Klose et al., 2014*; *Constantinides et al., 2014*; *Yang et al., 2015*). Given the many ways in which NK cells and ILC1s differ, ILC classification was recently revised so that NK cells and ILC1s are now considered distinct ILC lineages (*Vivier et al., 2018*).

Constructing clear definitions of NK cells and ILC1s is complicated, however, by their plasticity and heterogeneity upon stimulation, which can alter the features that discriminate between them under steady state conditions. For example, within the tumor microenvironment or upon in vitro culture with TGF-β, NK cells may downregulate Eomes and resemble ILC1s (*Gao et al., 2017*; *Cortez et al., 2016*). However, these cells are found only in the tumor microenvironment in vivo and it is not clear if they can persist in the absence of tumor. Moreover, NK cells and ILC1s become activated under similar circumstances, such as murine cytomegalovirus (MCMV) infection and tumorigenesis (*Weizman et al., 2017*; *Dadi et al., 2016*; *Yokoyama, 2013*). As it is currently difficult to assess NK cells independently of ILC1s, changes undergone by either population may be masked by the presence of the other. Independent examination of each population, especially in settings known to stimulate NK cells and ILC1s, is required to better understand these cells and their relationship to each other.

Both NK cells and ILC1s respond to infection with *Toxoplasma gondii*, an intracellular parasite that infects a third of the global population and is a natural pathogen in mice (*Montoya and Liesenfeld, 2004*). During acute infection, rapidly replicating tachyzoites disseminate systemically, triggering dendritic cells to produce IL-12, which subsequently stimulates NK cells to produce IFNγ and ILC1s to produce IFNγ and TNFα (*Klose et al., 2014*; *Goldszmid et al., 2012*). Tachyzoites eventually differentiate into slowly replicating bradyzoites, which are primarily encysted within cells of the central nervous system and skeletal muscle (*Yarovinsky, 2014*). As *T. gondii* infection elicits activation of both NK cells and ILC1s, herein we sought to investigate how NK cells and ILC1s respond to gain better insight into inflammation-induced changes. Indeed, we found that ILC1s become permanently heterogeneous after infection, largely owing to the surprising conversion of NK cells into ILC1-like cells.

## Results

### *T. gondii* infection results in expansion of ILC1-like cells

Following administration of anti-NK1.1, acute infection with the type II Prugniaud (Pru) strain of *T. gondii* resulted in increased parasite load and higher mortality rates as compared to isotype control-treated mice (*Figure 1—figure supplement 1A–C*), consistent with previous reports (*Goldszmid et al., 2007*). Since anti-NK1.1 affects both NK cells and ILC1s and both have been previously implicated in the immune response to *T. gondii* (*Goldszmid et al., 2012*; *Klose et al., 2014*), we sought to investigate how NK cells and ILC1s respond to infection. Here, we assessed these populations by following expression of Eomesodermin (Eomes) and CD49a among CD3⁻ CD19⁻ NK1.1⁺ NKp46⁺ cells, as NK cells express Eomes and not CD49a while ILC1s express CD49a but not Eomes at steady-state. In the uninfected spleen, the vast majority of NK1.1⁺ NKp46⁺ cells were NK cells (*Figure 1A*). There was a small population of spleen cells resembling ILC1s in uninfected mice, even though ILC1s are primarily found in other organs including the liver and small intestine and are generally tissue-resident (*Sojka et al., 2014a*; *Fuchs et al., 2013*), whereas cells in the spleen are generally circulating cells (*Gasteiger et al., 2015*; *Sojka et al., 2014a*; *Peng et al., 2013*). Interestingly, over the course of infection, NK cells decreased both as a proportion of NK1.1⁺ NKp46⁺ cells and in absolute number (*Figure 1A,B*). By contrast, there was an increase in cells resembling ILC1s that was clearly evident at 21-day post-infection (p.i.) (*Figure 1A,C*). Since ILC1 markers were established for ILC1s in uninfected mice, we have termed these cells resembling ILC1s as 'ILC1-like cells.' Notably,

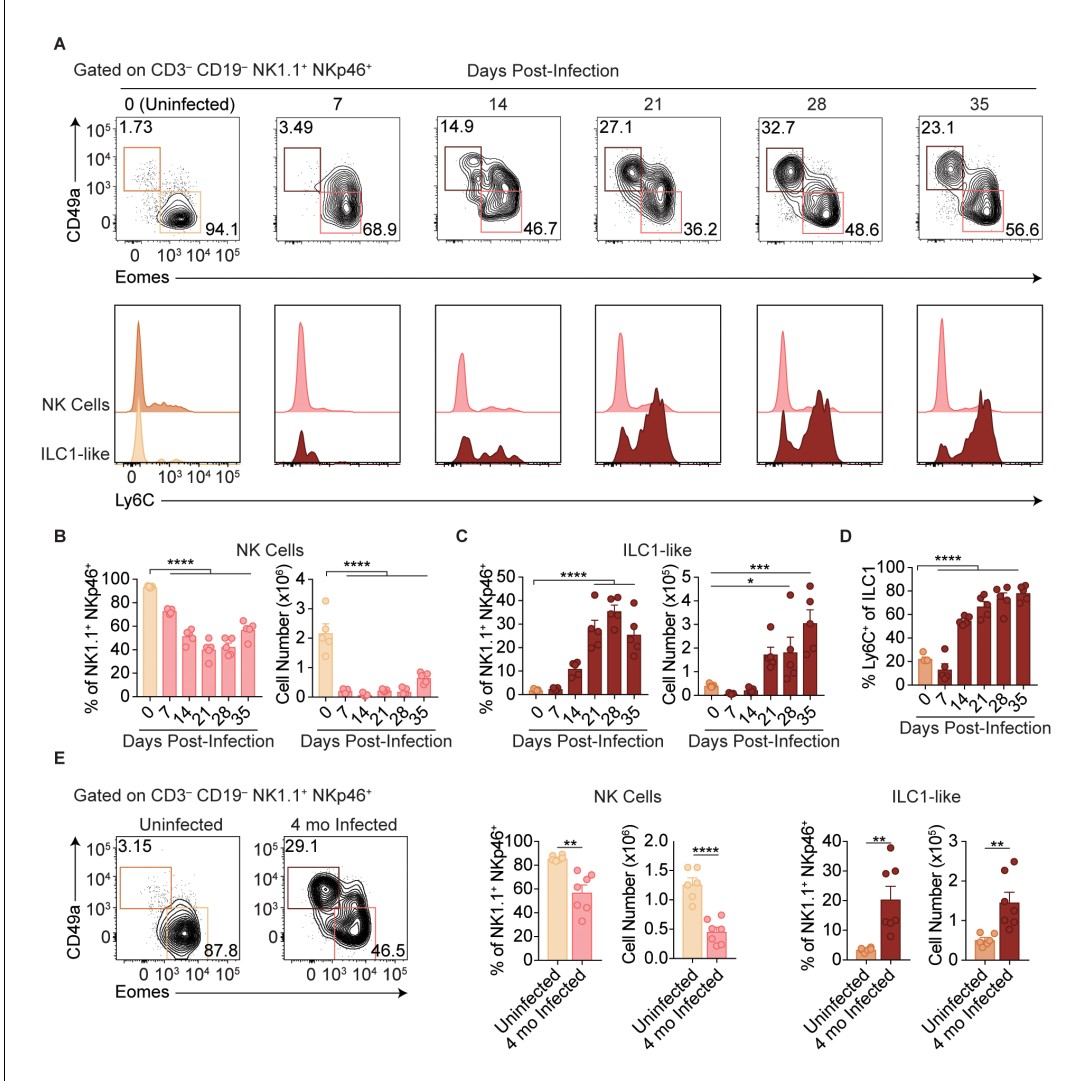

**Figure 1.** *T. gondii* infection results in expansion of ILC1-like cells. (A–E) Wild-type mice were infected by i.p. injection of 200 tachyzoites of the Prugniaud (Pru) strain of *T. gondii*. Splenocytes were analyzed at indicated time points p.i. (A) Representative flow cytometry plots for the analysis of NK cells (Eomes$^+$ CD49a$^-$) and ILC1-like cells (Eomes$^-$ CD49a$^+$), and Ly6C expression by NK cells and ILC1-like cells, at indicated time points p.i., among CD3$^-$ CD19$^-$ NK1.1$^+$ NKp46$^+$ cells. (B,C) Frequency and absolute number of NK cells (B) and ILC1-like cells (C) from spleens of uninfected mice and mice at indicated time points p.i. Numbers derived from gates indicated in (A), n = 5 mice. (D) Frequency of ILC1s that express Ly6C at indicated time points p.i., n = 5 mice. (E) Representative flow cytometry plots for the analysis of NK cells and ILC1-like cells, and their frequency and absolute number four mo p.i., n = 6 mice (uninfected) or n = 7 mice (infected). Mean + s.e.m (B–E); one-way ANOVA with Bonferroni correction (B–D); unpaired t-test (E); *p≤0.05, **p≤0.01, ***p≤0.001, ****p≤0.0001. Data are representative of 5 independent experiments.

DOI: https://doi.org/10.7554/eLife.47605.002

The following figure supplement is available for figure 1:

**Figure supplement 1.** Anti-NK1.1 depletion increases parasite burden and mortality in *T.gondii* infection.

DOI: https://doi.org/10.7554/eLife.47605.003

ILC1-like cells from infected mice mostly expressed Ly6C (*Figure 1A,D*), a marker that correlates with NK cell maturity (*Omi et al., 2014*) and is expressed by MCMV-induced memory NK cells (*Sun et al., 2012*) and is not expressed by the vast majority of ILC1-like cells present in the spleen under steady-state conditions. The expansion of splenic ILC1-like cells persisted for at least 4 months p.i. (*Figure 1E*).

## ILC1-like cell expansion persists in the absence of ongoing infection

Two major possibilities could account for the sustained expansion of ILC1-like cells. They may represent a response to ongoing parasite replication, as bradyzoites develop following acute infection with Pru *T. gondii* infection and parasite reactivation can occur. Alternatively, the increased ILC1-like cells might represent a permanent change that persists even after infection subsides. To test whether ongoing parasite replication plays a role, we infected mice with *T. gondii*, then suppressed growth of tachyzoites with sulfadiazine at various time points p.i. (*Eyles and Coleman, 1955*). Interestingly, infection-induced expansion of ILC1-like cells, as seen at 35 d.p.i, even in sulfadiazine-treated mice when treatment began 7 or 10 d.p.i., with cell numbers comparable to those of untreated, infected mice (*Figure 2A*). ILC1-like cell expansion also occurred following repeated administration of the *Cps1-1 T. gondii* strain (*Figure 2B*), which does not replicate in vivo and therefore does not persist in a chronic form (*Fox and Bzik, 2002*). Moreover, the number of ILC1-like cells

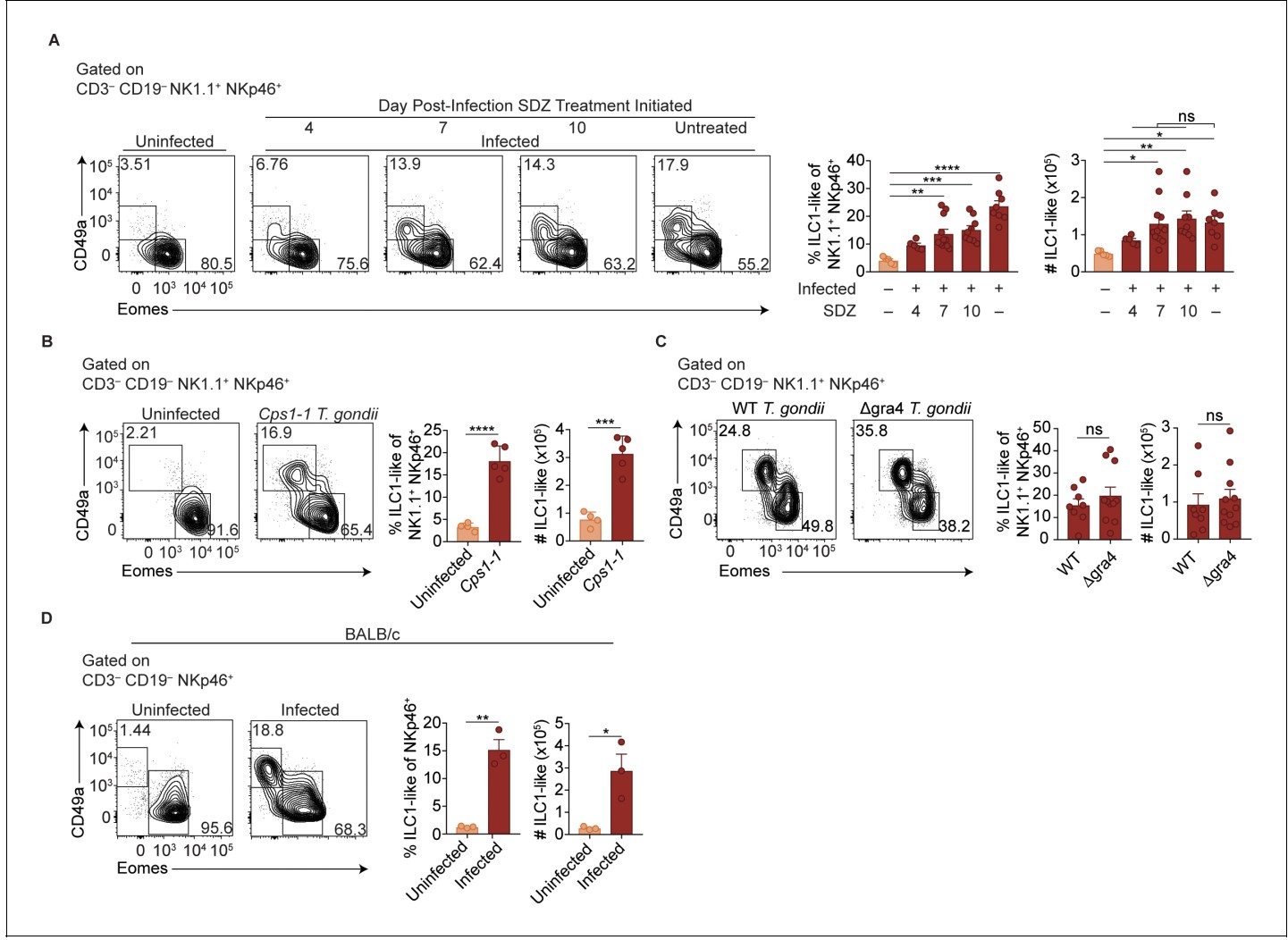

**Figure 2.** ILC1-like cell expansion persists in the absence of ongoing infection. (**A–D**) Representative flow cytometry plots for the analysis of NK cells (Eomes+ CD49a−) and ILC1-like cells (Eomes− CD49a+), and frequency and absolute number of ILC1-like cells in the spleen following (**A**) treatment with sulfadiazine (SDZ) beginning at indicated time points p.i. and maintained on sulfadiazine until 35 d p.i. when they were analyzed, n = 5–11 mice; (**B**) injection with 1 × 10^5 tachyzoites three times, two wk apart, and analyzed 42 d after initial injection, n = 4 mice (uninfected) or n = 5 mice (*Cps1-1*); (**C**) infection with PruΔku80 (WT) and PruΔku80Δgra4 (Δgra4) *T. gondii* analyzed 35 d.p.i., n = 8 mice (WT) or n = 10 mice (Δgra4); (**D**) infection of BALB/c mice 35 d.p.i., n = 3 mice. Mean + s.e.m (**A–D**); one-way ANOVA with Bonferroni correction (**B**); unpaired t-test (**B–D**); *ns* not significant, *p≤0.05, **p≤0.01, ***p≤0.001, ****p≤0.0001. Data are representative of three independent experiments.
DOI: https://doi.org/10.7554/eLife.47605.004

was similar after infection with a single inoculum of Δgra4 parasites, which are defective in cyst formation (*Fox et al., 2011*; *Jones et al., 2017*), as compared to the WT parental Pru strain (*Figure 2C*). BALB/c mice, which have fewer cysts during chronic *T. gondii* infection (*Suzuki et al., 1993*), also had increased numbers of ILC1-like cells after Pru infection (*Figure 2D*), also indicating that expanded ILC1-like cells were not confined to infected C57BL/6 mice. In summary, we found that *T. gondii* infection induced expansion of ILC1-like cells, which persisted independent of ongoing parasite replication, suggesting a permanent change; in subsequent experiments, we further examined the impact of infection with the Pru strain of *T. gondii*.

## Expansion of ILC1-like cells among circulating cells

ILC1s have previously been established as being tissue-resident under steady state conditions (*Gasteiger et al., 2015*; *Sojka et al., 2014a*; *Peng et al., 2013*). Surprisingly, however, we found that infected mice displayed ILC1-like cell expansion even in the blood, and a highly vascularized organ, that is the lung, indicating that at least some of these cells circulate (*Figure 3A,B*). Consistent with our previous findings in the spleen (*Figure 1A,D*), ILC1-like cells in the blood and lungs of infected mice also expressed Ly6C (*Figure 3—figure supplement 1A*). However, there was no increase of ILC1-like cells in the mesenteric lymph node, brain, peritoneum, bone marrow, uterus, or salivary gland (*Figure 3C*, *Figure 3—figure supplement 1B*). Nonetheless, these data indicated that *T. gondii* infection induced ILC1-like cell expansion, even in the circulation.

Parabiosis experiments could provide supportive evidence of circulating versus tissue-resident cells, but the possible reactivation and spread of *T. gondii* following surgery, as seen clinically (*Bosch-Driessen et al., 2002*), could confound analysis. Instead, we utilized *Eomes*-GFP reporter mice to assess how ILC1-like cells and NK cells from spleens of infected mice behave following transfer into uninfected mice (*Daussy et al., 2014*). We isolated NK cells and ILC1-like cells from congenically distinct *Eomes*-GFP mice 35 d.p.i. for adoptive transfer into naive *Rag2^-/-^Il2 rg^-/-^* mice (*Figure 4A*). When we assessed the recipient mice 24 days later, we detected both transferred populations in the spleen and liver, though there was a much smaller percentage of transferred NK cells in the liver (*Figure 4B*). Although NK cells upregulated CD49a upon transfer into *Rag2^-/-^Il2 rg^-/-^* mice, as previously reported (*Gao et al., 2017*), Eomes expression remained high in most transferred NK cells, compared to most transferred ILC1-like cells which remained Eomes⁻ (*Figure 4C*).

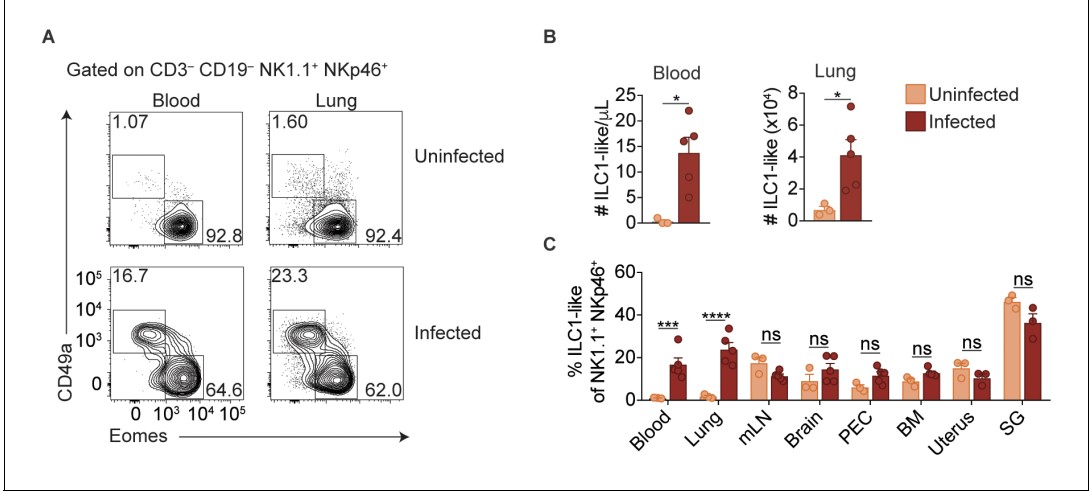

**Figure 3.** Expansion of ILC1-like cells among circulating cells. (**A**) Representative flow cytometry plots for the analysis of NK cells and ILC1-like cells in the blood and lung of uninfected and d35-infected mice. (**B**) Absolute number of ILC1-like cells in the blood and lung of uninfected and d35 infected mice. (**C**) Frequency of ILC1-like cells among NK1.1⁺ NKp46⁺ cells in indicated organ at 35 d.p.i., *n* = 3 mice (uninfected) or *n* = 5 mice (infected). Mean + s.e.m (**A,C**); unpaired *t*-test; *ns* not significant, *p≤0.05, ***p≤0.001, ****p≤0.0001. Data are representative of three independent experiments.
DOI: https://doi.org/10.7554/eLife.47605.005
The following figure supplement is available for figure 3:

**Figure supplement 1.** ILC1s expand at additional sites beyond the spleen.
DOI: https://doi.org/10.7554/eLife.47605.006

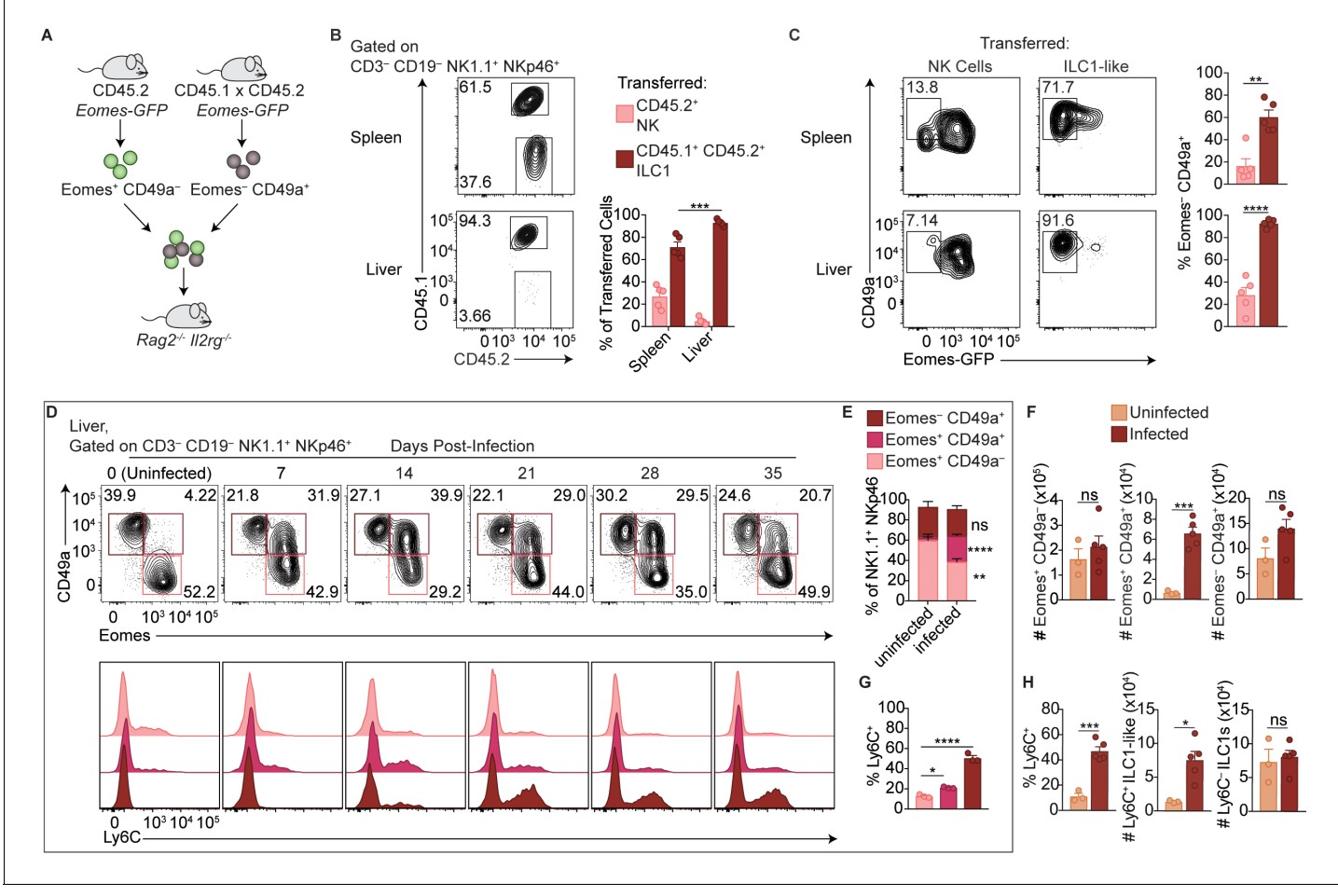

**Figure 4.** Delineation of NK cells, ILC1s, and ILC1-like cells in the liver. (A) Overview of competitive transfer experiment. At 35 d.p.i., Eomes-GFP⁺ CD49a⁻ and Eomes-GFP⁻ CD49a⁺ cells were sorted from the spleens of CD45.2 and CD45.1 x CD45.2 *Eomes*-GFP reporter mice, respectively. Cells were combined in a 1:1 ratio and intravenously injected into *Rag2⁻/⁻ Il2rg⁻/⁻* mice. 24 days later, spleens and livers of recipients were assessed for transferred cells. (B) Representative flow cytometry plots of CD45.1 and CD45.2 expression by transferred cells in the spleen and liver of *Rag2⁻/⁻ Il2rg⁻/⁻* mice, and frequency of transferred NK cells (CD45.2⁺) and ILC1s (CD45.1⁺ CD45.2⁺) 24 days post-transfer, as described in (A), n = 5 mice. (C) Representative flow cytometry plots of Eomes-GFP and CD49a expression by transferred NK cells and ILC1-like cells in the spleen (top) and liver (bottom) of *Rag2⁻/⁻ Il2rg⁻/⁻* mice, and frequency of Eomes⁻ CD49a⁺ 24 days post-transfer, as described in (A) n = 5 mice. (D) Representative flow cytometry plots for the analysis of CD49a and Eomes among CD3⁻ CD19⁻ NK1.1⁺ NKp46⁺ cells, and Ly6C expression by Eomes⁺ CD49a⁻, Eomes⁺ CD49a⁺, and Eomes⁻ CD49a⁺ cells in the liver at indicated time points p.i. (E,F) Frequency and absolute number of CD3⁻ CD19⁻ NK1.1⁺ NKp46⁺ cells that are Eomes⁺ CD49a⁻, Eomes⁺ CD49a⁺, and Eomes⁻ CD49a⁺ in the livers of uninfected and d35-infected mice, n = 3 mice (uninfected) or n = 5 mice (infected). (G) Frequency of Ly6C⁺ cells among CD3⁻ CD19⁻ NK1.1⁺ NKp46⁺ Eomes⁺ CD49a⁻, Eomes⁺ CD49a⁺, and Eomes⁻ CD49a⁺ cells in the livers of d35-infected mice, n = 3 mice. (H) Frequency of CD3⁻ CD19⁻ NK1.1⁺ NKp46⁺ cells that are Eomes⁻ CD49a⁺ (ILC1 and ILC1-like) and express Ly6C, and absolute number of Ly6C⁺ ILC1-like cells and Ly6C⁻ ILC1s from livers of uninfected and d35-infected mice, n = 3 mice (uninfected) or n = 5 (infected). Mean + s.e.m. (B,C,E–H) unpaired t-test (B,C,E,F,H) one-way ANOVA with Bonferroni correction (G); *ns* not significant, *p≤0.05, **p≤0.01, ***p≤0.001, ****p≤0.0001. Data are representative of 3 independent experiments.

DOI: https://doi.org/10.7554/eLife.47605.007

Interestingly, following adoptive transfer of ILC1-like cells, some Eomes⁻ cells recovered from the spleen expressed Eomes (*Figure 4C*), suggesting that in certain contexts, Eomes⁻ ILC1-like cells may give rise to Eomes⁺ cells, similar to what has been shown previously for ILC1s (*Gordon et al., 2012*; *Takeda et al., 2005*). Therefore, in the setting of homeostatic expansion, most transferred NK cells and ILC1-like cells from infected mice persist with generally stable Eomes expression, and transferred ILC1-like cells from infected mice circulate, providing complementary evidence, in addition to above (*Figure 2A–D*), that they persist in the absence of ongoing infection.

Interestingly, however, the ratio of transferred ILC1-like cells to NK cells was higher in the liver than in the spleen, with ILC1-like cells comprising nearly all transferred cells in the liver (*Figure 4B*). This suggested that the ILC1-like cells from infected mice may not uniformly circulate and rather they might preferentially migrate to the liver despite originating in the spleen. Alternatively, the adoptive transfer studies may be revealing properties of heterogeneous populations of cells. To further evaluate these issues, we studied the liver, which contains a large population of ILC1s in uninfected mice (*Sojka et al., 2014a*; *Peng et al., 2013*), in more detail directly following primary infection. Infection decreased the frequency of liver NK cells among NK1.1$^+$ NKp46$^+$ cells, although the total NK cell number did not change (*Figure 4D–F*). Notably, we observed that infection resulted in a striking population of liver Eomes$^+$ CD49a$^+$ cells (*Figure 4D–F*). The frequency and absolute number of ILC1s also did not change (*Figure 4D–F*). We examined Ly6C expression by Eomes$^+$ CD49a$^-$, Eomes$^+$ CD49a$^+$, and Eomes$^-$ CD49a$^+$ cells in the infected liver and found that Ly6C was expressed most highly by Eomes$^-$ CD49a$^+$ cells after infection (*Figure 4D,G*), and to a lesser extent by Eomes$^+$ CD49a$^+$ cells. Interestingly, Ly6C was expressed at a higher percentage by cells with an ILC1 phenotype (Eomes$^-$ CD49a$^+$) in the infected spleen (~80%, *Figure 1D*) than in the infected liver, where only half of such cells expressed Ly6C (*Figure 4H*), suggesting heterogeneity among liver cells resembling ILC1s. Accordingly, the number of liver Ly6C$^+$ ILC1-like cells (Eomes$^-$ CD49a$^+$) was increased relative to uninfected mice, whereas the number of Ly6C$^-$ ILC1s (Eomes$^-$ CD49a$^+$) remained comparable (*Figure 4H*), suggesting that at 35 d.p.i. Ly6C expression may distinguish between steady-state ILC1s and ILC1-like cells induced by infection. Taken together, the data suggest that in infected mice, steady-state ILC1s are maintained while an additional ILC1-like subset arises and circulates.

## ILC1-like cells are distinct from NK cells and ILC1s

Ly6C is expressed by ILC1-like cells in the spleen, blood, and liver of infected mice, suggesting that *T. gondii* induces ILC1-like cells that are rare under steady-state conditions. To assess the possibility of a novel ILC1-like subset, we first performed more detailed comparisons of NK cells and ILC1-like cells in the spleens of uninfected and infected mice. As circulatory capacity was specific to ILC1-like cells in infected mice, we examined expression of molecules typically associated with ILC1 tissue-residency (*Sojka et al., 2014a*; *Daussy et al., 2014*). ILC1-like cells in both uninfected and infected mice did not express CD62L and expressed higher levels of PECAM-1 and CXCR6 compared to NK cells (*Figure 5A*). However, infection resulted in NK cells and ILC1-like cells gaining expression of CCR8 and CX$_3$CR1 compared to NK cells and ILC1s in uninfected mice. We also assessed expression of DNAM-1 and KLRG1, which correlate with the maturational state of NK cells (*Goh and Huntington, 2017*). We found that ILC1-like cells from infected mice expressed DNAM-1 more highly than NK cells, and KLRG1 at higher levels than NK cell or ILC1 populations (*Figure 5B*). We also observed that *T. gondii*-induced ILC1-like cells were unique in their expression of Neuropilin-1, a cell surface protein involved in activation of regulatory T cells (*Sarris et al., 2008*), but not previously described in NK cells or ILC1s.

NK cell development is classically described as a stepwise maturation process through CD27$^-$ CD11b$^-$, CD27$^+$ CD11b$^-$, CD27$^+$ CD11b$^+$, and CD27$^-$ CD11b$^+$ stages (*Chiossone et al., 2009*), while ILC1s do not undergo these stages and are primarily CD27$^+$ CD11b$^-$ (*Sojka et al., 2014a*). In the spleen, we found that ILC1s from uninfected mice were predominantly CD27$^+$ CD11b$^-$ and NK cells from uninfected and infected mice encompassed all subsets (*Figure 5—figure supplement 1A*), as previously shown. However, ILC1-like cells from infected mice were mainly CD27$^-$ CD11b$^+$ (*Figure 5—figure supplement 1A*). With respect to NK cell receptor expression, ILC1-like cells from infected mice resembled ILC1s from uninfected mice, expressing NKG2A and low frequencies of Ly49 receptors (*Figure 5—figure supplement 1B*). Regarding cytokine production, NK cells and ILC1s in both uninfected and infected mice produced IFNγ in response to stimulation (*Figure 5C*), a hallmark of both cell types (*Sojka et al., 2014b*). Similarly, ILC1-like cells also produced IFNγ. However, only ILC1s from uninfected mice produced TNFα (*Figure 5D*). Similar results were obtained from studies of ILC1-like cells in blood of infected mice (*Figure 5—figure supplement 1C,D*). Thus, the spleens and blood of infected mice contained a novel ILC1-like subset.

Studies of the infected liver solidified the notion that infection induced a distinct ILC1-like subpopulation. In the liver, we separately assessed the Ly6C$^-$ ILC1s that are present in uninfected mice, as well as both Ly6C$^-$ ILC1s and Ly6C$^+$ ILC1-like cells in infected mice (*Figure 5—figure supplement*

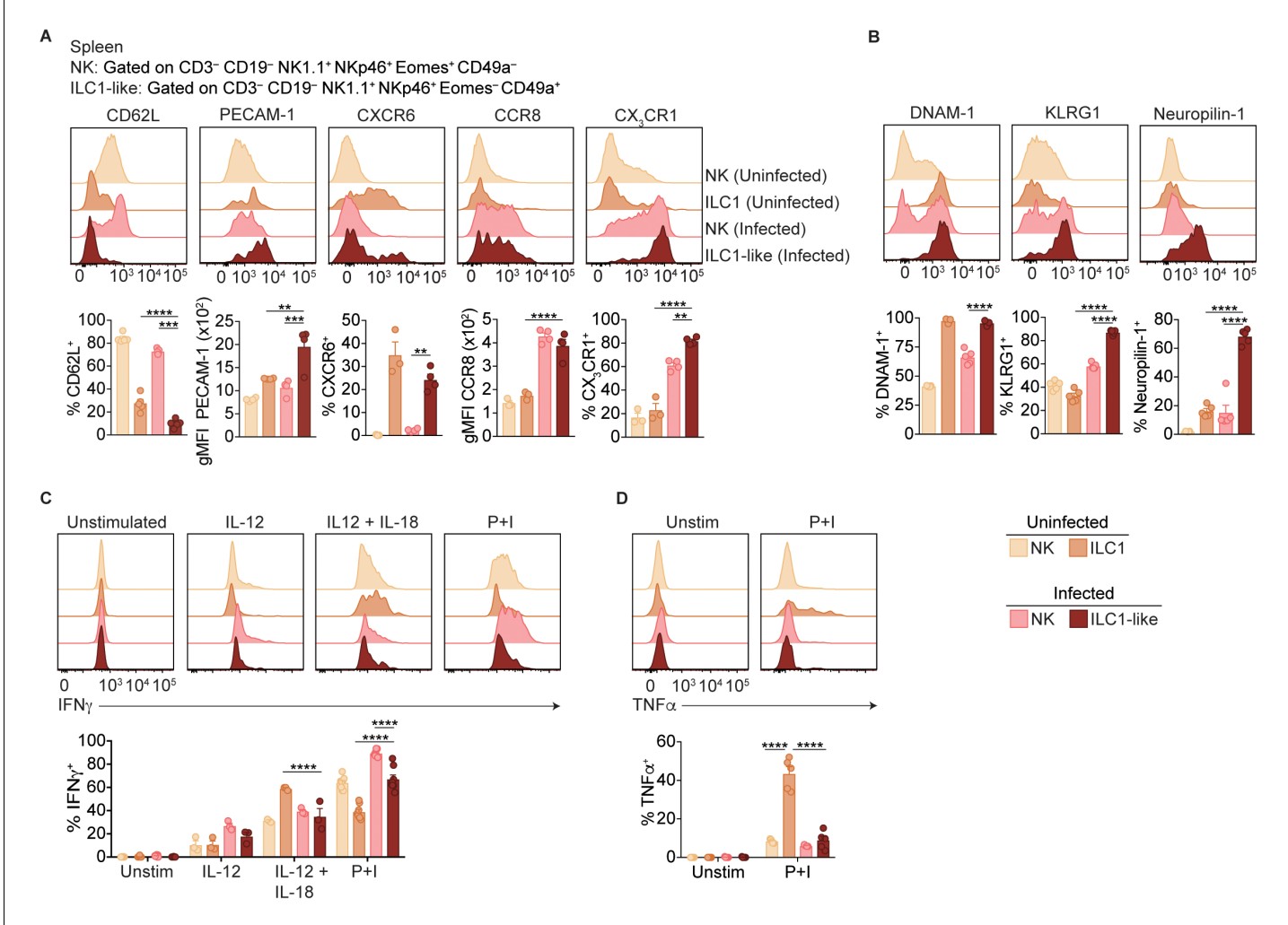

**Figure 5.** ILC1-like cells are distinct from NK cells and ILC1s. (A,B) Representative flow cytometry plots showing expression of indicated markers by indicated cells from spleens of uninfected and d35-infected mice, and frequency of each cell population that expresses each marker, n = 3 mice (A) uninfected) or n = 5 mice (A) infected and B). (C,D) Representative flow cytometry plots and frequency of indicated cells in uninfected and d35-infected mice after 4 hr culture with indicated stimulus that contain intracellular IFNγ (C) or TNFα (D), n = 5. Mean + s.e.m. (A–D) one-way ANOVA with Bonferroni correction (A–D); *p≤0.05, **p≤0.01, ***p≤0.001, ****p≤0.0001, in comparisons of ILC1-like cells from infected mice with the NK cells in infected mice, and the ILC1s in uninfected mice. When these comparisons were not significant, the comparison is not shown. Data are representative of three independent experiments.

DOI: https://doi.org/10.7554/eLife.47605.008

The following figure supplements are available for figure 5:

**Figure supplement 1.** Phenotypic comparison of NK cells, ILC1s, and ILC1-like cells in the spleen and blood.
DOI: https://doi.org/10.7554/eLife.47605.009

**Figure supplement 2.** Ly6C expression distinguishes between ILC1 subpopulations in the liver.
DOI: https://doi.org/10.7554/eLife.47605.010

*2A*). Ly6C$^+$ ILC1-like cells expressed higher levels of CX$_3$CR1, Neuropilin-1, KLRG1, and CD11b, lower levels of CXCR6 and CD27, and produced less TNFα than Ly6C$^-$ ILC1s in uninfected and infected mice, which were similar (*Figure 5—figure supplement 2B–D*). Therefore, the Ly6C$^+$ ILC1-like cells in the liver therefore resembled those in the infected spleen (*Figure 5A–D*, *Figure 5—figure supplement 1A,B*), suggesting that while ILC1s in uninfected mice and ILC1-like cells in infected

mice share the Eomes⁻ CD49a⁺ phenotype, they can be distinguished by marker expression and cytokine production profile.

## *T. gondii* infection induces heterogeneity of NK cells and ILC1s

ILC1-like cells from spleens of infected mice expressed high levels of Tbet (*Figure 6A*), which was previously identified as a requirement for ILC1 development (*Daussy et al., 2014*; *Sojka et al., 2014a*). We therefore hypothesized that Tbet may be required for ILC1-like cell expansion after infection. As *Tbx21*⁻/⁻ mice succumb to *T. gondii* infection (*Harms Pritchard et al., 2015*), we studied WT:*Tbx21*⁻/⁻ mixed bone marrow chimeras. In spleens of both uninfected and infected chimeras, NK cells were present though they were biased towards WT origin (*Figure 6—figure supplement 1A*), consistent with a requirement for Tbet to complete NK cell maturation (*Gordon et al., 2012*). However, no ILC1-like cells (Eomes⁻ CD49a⁺) were found of *Tbx21*⁻/⁻ origin in infected WT:*Tbx21*⁻/⁻ chimeras, as with ILC1s in naive chimeras (*Figure 6B*). Thus, splenic ILC1-like cells are Tbet-dependent.

In the livers of infected WT:*Tbx21*⁻/⁻ chimeras, the picture was more complex. There were no ILC1s (Eomes⁻ CD49a⁺) of *Tbx21*⁻/⁻ origin in the infected liver, as with infected spleens (*Figure 6B*). However, the livers of infected WT:*Tbx21*⁻/⁻ chimeras displayed an accumulation of *Tbx21*⁻/⁻ Eomes⁺ CD49a⁺ cells that were not present in the spleen (*Figure 6B*). Eomes⁺ CD49a⁺ cells were also present among counterpart WT cells, though at lower frequencies. These data suggested that ILC1-like cells in infected mice may be derived from Eomes⁺ cells, that is, NK cells.

To better assess the heterogeneity induced by *T. gondii* infection and the role of Tbet in this process, we performed single-cell RNA-seq (scRNA-seq) on sorted WT (CD45.1⁺) or *Tbx21*⁻/⁻ (CD45.2⁺) NK1.1⁺ NKp46⁺ cells from uninfected and d35-infected WT:*Tbx21*⁻/⁻ chimeras. We chose to examine the liver for this analysis for several reasons: 1) infection appears to induce a novel ILC1-like subset that is common to the liver and spleen; 2) the liver contains ILC1s under steady-state, allowing for direct comparison of steady-state ILC1s and *T. gondii*-induced ILC1-like cells; and 3) the infected liver uniquely contains a *Tbx21*⁻/⁻ Eomes⁺ CD49a⁺ population. Using t-distributed stochastic neighbor embedding (t-SNE) analysis, we grouped the cells into 17 clusters, C1-C17 (*Figure 6C*). In the uninfected WT sample, C1 was the largest cluster, whereas C1 was diminished in the infected WT and *Tbx21*⁻/⁻ samples (*Figure 6D,E*). C1 cells expressed *Eomes* and *Itgam* but did not express *Cd27* or *Itga1* (*Figure 6F,G*), consistent with C1 being mature NK cells, which are indeed the predominant NK1.1⁺ NKp46⁺ cells present in the steady-state liver (*Figure 6—figure supplement 1B–D*). C3 was the second largest cluster in the uninfected WT sample, and C3 cells did not express *Eomes* or *Ly6c2*, expressed *Itga1*, *Cd27*, and *Cxcr6*, and were absent in *Tbx21*⁻/⁻ samples, indicating that C3 comprised ILC1s (*Figure 5F,G*, *Figure 6—figure supplement 1D*).

Two closely related clusters, C10 and C11, were unique to the infected WT sample (*Figure 6D,E*). Cells in C10 and C11 expressed *Itga1*, *Ly6c2*, and *Klrg1*, and did not express *Eomes* (*Figure 6F,G*). This specificity to infection, Tbet-dependence, and gene expression identified these cells as *T. gondii*-induced ILC1-like cells. Interestingly, these two clusters indicated heterogeneity even within the *T. gondii*-induced ILC1-like population. The t-SNE analysis placed C10 closer to mature NK cells (C1) and C11 closer to ILC1s (C3). Comparison of C10 and C11 revealed that C10 expressed signature NK cell genes at higher levels (*Robinette et al., 2015*), whereas C11 expressed higher levels of signature ILC1 genes (*Figure 6—figure supplement 1E*). Interestingly, C10 also highly expressed *Cx3cr1* whereas C11 did not (*Figure 6F,G*). We corroborated these findings with flow cytometric analysis that showed that Ly6C⁺ ILC1-like cells in the infected liver contained both a CX₃CR1⁺ CXCR6⁻ population that expressed higher levels of Zeb2, mirroring the C10 phenotype, and also a CX₃CR1⁻ CXCR6⁺ population that mirrors C11 (*Figure 6—figure supplement 1F,G*). Furthermore, C10 expressed lower levels of *Il7ra* than C3 and C11 (*Figure 6F*). As IL-7 receptor signaling is required for ILC1 development but not for NK cells (*Klose et al., 2014*), this finding is suggestive of discrete developmental origins of C10 and C11, which we further discuss below. C5 was also increased in the infected samples, and these cells co-expressed *Eomes* and *Itga1*, and were especially prominent in the infected *Tbx21*⁻/⁻ sample (*Figure 6E,G*), suggesting that this cluster includes the Eomes⁺ CD49a⁺ cells that arose in the liver after infection (*Figures 4E* and *6B*). In summary, scRNA-seq analysis revealed that *T. gondii* infection induces heterogeneous NK cell and ILC1 populations, including two distinct ILC1-like subpopulations that are unique to infected mice.

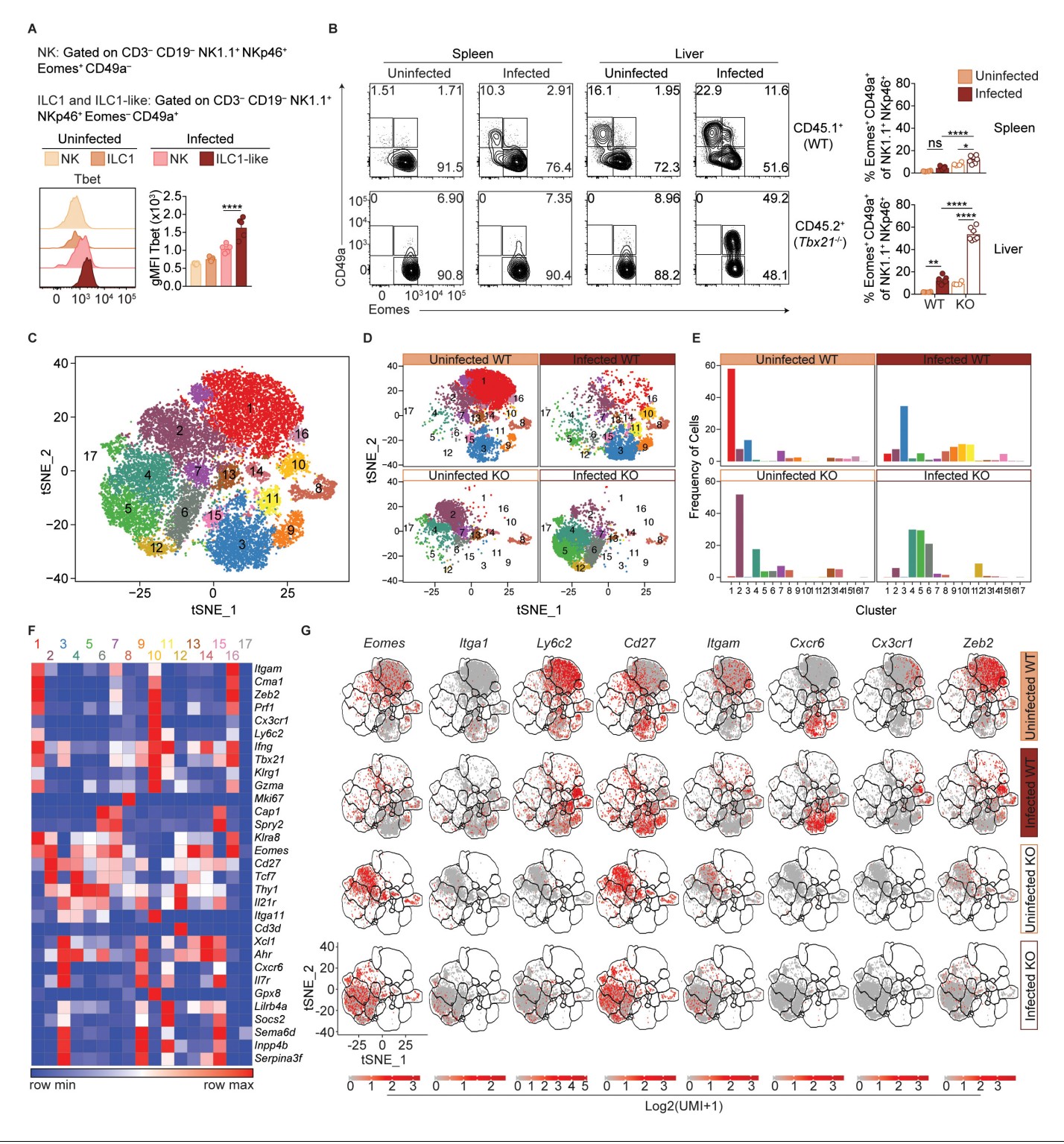

**Figure 6.** *T.gondii* infection induces heterogeneity of NK cells and ILC1s. (**A**) Representative flow cytometry plots showing Tbet expression by indicated cells from spleens of uninfected and d35-infected mice, and gMFI of Tbet expression by each population, *n* = 3 mice (uninfected) or *n* = 5 mice (infected). (**B**) Representative flow cytometry plots for the CD49a and Eomes expression among CD3⁻ CD19⁻ NK1.1⁺ NKp46⁺ cells of CD45.1-derived (WT) or CD45.2-derived (*Tbx21⁻/⁻*) cells from spleens and livers of uninfected and d35-infected WT:*Tbx21⁻/⁻* bone marrow chimeras, and frequency of Eomes⁺ CD49a⁺ cells from spleen and liver, *n* = 4 mice (uninfected) or *n* = 7 mice (infected). In right panel, KO cells indicated by open bars. (**C**) Biaxial t-SNE analysis of cells pooled across all samples. Points represent individual cells, colors denote different clusters. (**D**) Biaxial t-SNE analysis of cells in

*Figure 6 continued on next page*

*Figure 6 continued*

individual samples. (**E**) Fraction of cells in C1-C17 within each sample. (**F**) Heatmap of average expression of select genes that are characteristic of C1-C17. (**G**) Expression of indicated genes by cells in each sample. Mean + s.e.m. (**A,B**); one-way ANOVA with Bonferroni correction (**A,B**); *ns* not significant *$p \leq 0.05$, **$p \leq 0.01$, ****$p \leq 0.0001$; data are representative of three independent experiments (**A–C**).

DOI: https://doi.org/10.7554/eLife.47605.011

The following figure supplement is available for figure 6:

**Figure supplement 1.** Single-cell RNA-seq reveals extensive NK cell and ILC1 heterogeneity in the liver.

DOI: https://doi.org/10.7554/eLife.47605.012

## NK cells downregulate Eomes during *T. gondii* infection

scRNA-seq analysis revealed heterogeneity of NK1.1$^+$ NKp46$^+$ cells and showed that some *T. gondii*-induced ILC1-like cells have core NK cell traits. To assess if NK cell conversion into ILC1-like cells might account for this population by means of Eomes downregulation, we examined *Ncr1*$^{iCre}$ *Eomes*$^{f/f}$ (*Eomes* conditional KO or cKO) mice, in which NK cells are absent owing to their requirement for Eomes, as shown by a drastic reduction of NK1.1$^+$ NKp46$^+$ cells (*Figure 7—figure supplement 1A*). Meanwhile, the number of CD49a$^+$ ILC1s is unaffected while the frequency of these cells among CD3$^-$ CD19$^-$ NK1.1$^+$ NKp46$^+$ cells is increased due to the absence of NK cells (*Figure 7A,B*) (*Gordon et al., 2012*; *Pikovskaya et al., 2016*). Following *T. gondii* infection, the numbers of CD49a$^+$ and CD49a$^+$ Ly6C$^+$ cells in the spleen were significantly reduced in *Eomes* cKO compared to *Eomes*$^{f/f}$ controls (*Figure 7A,B*), showing that expansion of these cells requires Eomes. However, while the overall number of CD49a$^+$ cells did not change between uninfected and infected *Eomes* cKO mice, the number of CD49a$^+$ Ly6C$^+$ cells increased slightly (*Figure 7B*), and a greater frequency of the CD49a$^+$ cells expressed Ly6C (*Figure 7C*), showing that mild ILC1 expansion occurs in the absence of NK cells (indicated by bracket). However, approximately three times as many CD49a$^+$ Ly6C$^+$ cells were derived from Eomes-dependent cells, that is, NK cells (*Figure 7B*, indicated by bracket), suggesting Eomes expression by NK cells contributes to the development of most ILC1-like cells.

The role of Eomes-expressing NK cells in formation of *T. gondii*-induced ILC1-like cells could be either cell-intrinsic or cell-extrinsic. Although an Eomes fate-mapping mouse, that is, *Eomes*$^{Cre}$, could be used to identify a cell-intrinsic role for Eomes, this was not possible due to the ubiquitous expression of Eomes during development (data not shown). Another approach would be to examine Eomes downregulation following adoptive transfer of congenic NK cells prior to infection but this was hindered due to the inability to find transferred cells after 14 d.p.i. (data not shown), possibly due to the high turnover of cells during infection. Rather, we used mixed bone marrow chimeras to examine a cell-intrinsic or cell-extrinsic requirement for Eomes during ILC1-like cell expansion. In uninfected mixed bone marrow chimeras reconstituted with WT cells and either *Eomes*$^{f/f}$ or *Eomes* cKO cells, the contributions of *Eomes*$^{f/f}$ and *Eomes* cKO cells to CD49a$^+$ cells were comparable, consistent with CD49a$^+$ cells being Eomes-independent ILC1s (*Figure 7D*). This degree of chimerism did not change upon infection of WT:*Eomes*$^{f/f}$ chimeras. However, infection of WT:*Eomes* cKO chimeras resulted in a significant reduction of CD49a$^+$ cells of *Eomes* cKO origin. Although there were a few CD49a$^+$ ILC1s of *Eomes* cKO origin, consistent with Eomes-independent ILC1 origin, *T. gondii* infection resulted in ILC1-like cells that were mostly dependent on a cell-intrinsic effect of Eomes in NKp46-expressing cells, that is, NK cells. Therefore most *T. gondii*-induced ILC1-like cells are ex-NK cells.

We postulated that STAT4 may play a role in the expansion of ILC1-like cells because STAT4 and IL-12 signaling is critical for NK cell and ILC1 activation during *T. gondii* infection (*Hunter et al., 1994*; *Cai et al., 2000*; *Klose et al., 2014*). To test the relevance of this signaling in vivo, we infected WT:*Stat4*$^{-/-}$ mixed bone marrow chimeras. Indeed, STAT4-deficient ILC1-like cells were diminished following infection, while chimerism of NK cells and ILC1s remained equal between uninfected and infected chimeric mice (*Figure 7e*), suggesting that *T. gondii*-induced ILC1-like cells were dependent on STAT4. Additionally, in vitro culture of splenocytes from infected mice resulted in an increased frequency of ILC1-like cells following culture in IL-2 and IL-12, compared to culture in IL-2 alone (*Figure 7—figure supplement 1B*), suggesting that IL-12 may contribute to ILC1-like cell

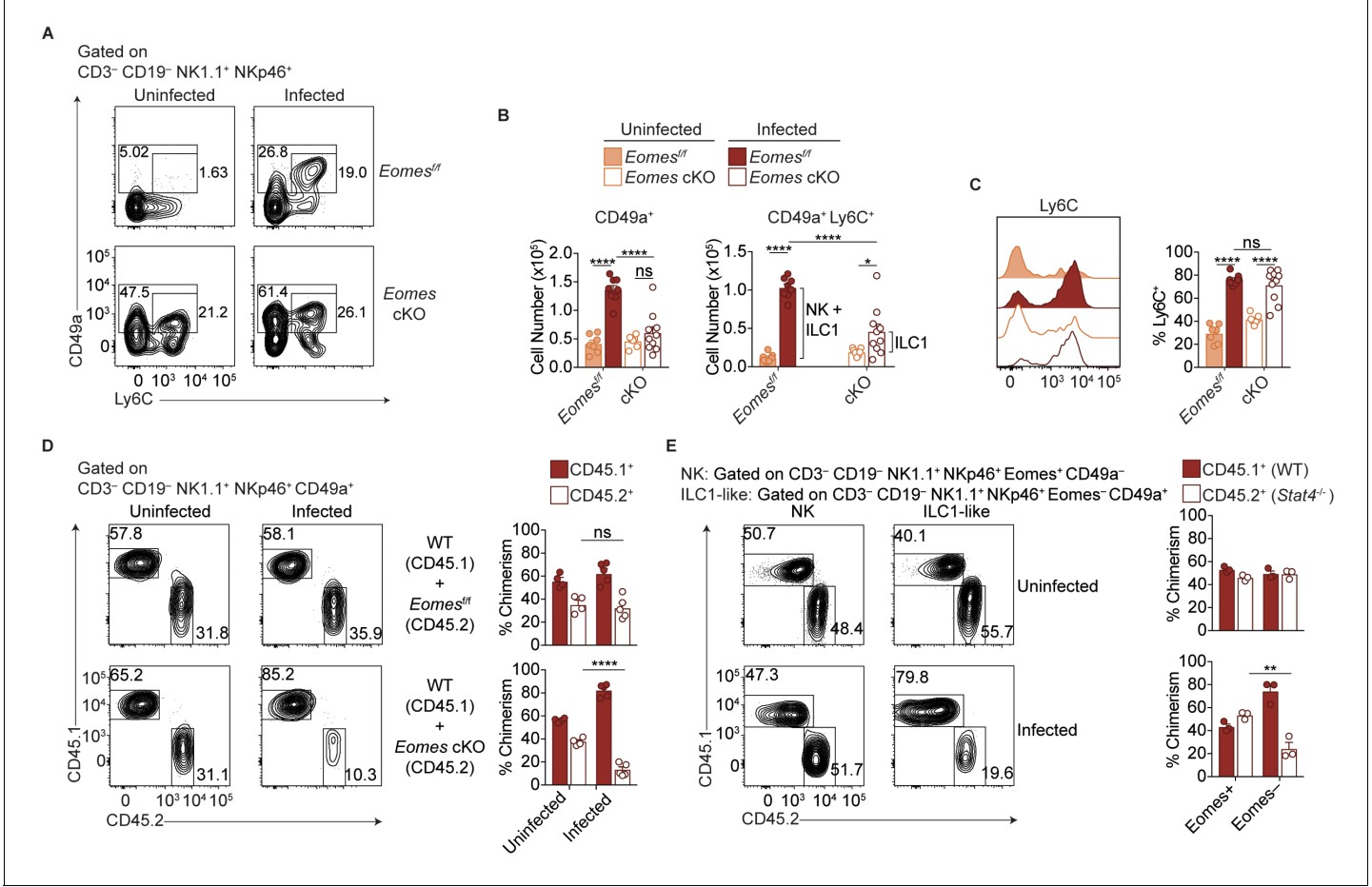

**Figure 7.** Most ILC1-like cells are derived from NK cells during *T.gondii* infection. (**A**) Representative flow cytometry plots for the analysis of ILC1s (CD3⁻ CD19⁻ NK1.1⁺ NKp46⁺ CD49a⁺) or ILC1-like cells (CD3⁻ CD19⁻ NK1.1⁺ NKp46⁺ CD49a⁺ Ly6C⁺), and (**B**) absolute number of these cells in the spleens of *Eomes*^f/f and *Eomes* cKO mice that are uninfected and at 35 d.p.i., *n* = 7 mice (uninfected *Eomes*^f/f), *n* = 8 mice (infected *Eomes*^f/f), *n* = 7 mice (uninfected cKO), or *n* = 9 mice (infected cKO). The increased number of CD49a⁺ Ly6C⁺ cells in infected versus uninfected *Eomes*^f/f mice represent both NK cells and ILC1 origin, whereas in cKO mice, the increased number represents ILC1 origin only, as indicated by the brackets. (**C**) Representative flow cytometry plot showing expression of Ly6C and frequency of ILC1s that express Ly6C, *n* = 7 mice (uninfected *Eomes*^f/f), *n* = 8 mice (infected *Eomes*^f/f), *n* = 7 mice (uninfected cKO), or *n* = 9 mice (infected cKO). (**D**) Representative flow cytometry plots for the analysis of chimerism in the spleens of uninfected and d35-infected WT:*Eomes*^f/f and WT:*Eomes* cKO bone marrow chimeras, and frequency of ILC1s that are derived from WT, *Eomes*^f/f, or *Eomes* cKO bone marrow, *n* = 4 mice (uninfected) or *n* = 5 (infected). (**E**) Representative flow cytometry plots for the analysis of chimerism in the spleens of uninfected and d35-infected WT:*Stat4*^-/- bone marrow chimeras, and frequency of NK cells and ILC1-like cells that are derived from WT or *Stat4*^-/- bone marrow, *n* = 3 mice. Mean + s.e.m. (**B–D**); one-way ANOVA with Bonferroni correction (**B–D**); *ns* not significant *p≤0.05, **p≤0.01, ****p≤0.0001. Data are representative of three independent experiments.

DOI: https://doi.org/10.7554/eLife.47605.013

The following figure supplement is available for figure 7:

**Figure supplement 1.** IL-12 contributes to Eomes downregulation.
DOI: https://doi.org/10.7554/eLife.47605.014

expansion. However, culture of purified NK cells from infected mice did not result in Eomes downregulation (***Figure 7—figure supplement 1C***), showing there may be IL-12 cell-extrinsic effects involved in this conversion. Notably, *T. gondii*-induced ILC1-like cells stably maintained their phenotype after 2 days in culture (***Figure 7—figure supplement 1C***). Taken together, these results suggest that STAT4 and IL-12 contribute to ILC1-like cell expansion in the context of *T. gondii* infection, with contributions from additional factors as well.

## Eomes downregulation within NK cells accompanies extensive transcriptional and epigenomic changes

Our scRNA-seq data suggested that *T. gondii* infection drastically rewires NK cells and ILC1s to induce permanent changes. To provide confirmatory evidence, we performed bulk RNA sequencing (RNA-seq) to inform epigenomic analyses, even though we realize that these approaches lose single cell resolution. For this analysis, we used splenocytes, sorted into NK cells (Ly6C$^-$ CD49a$^-$) from uninfected and infected mice, and ILC1-like cells (Ly6C$^+$ CD49a$^+$) from infected mice, which were accurately identified Eomes$^+$ and Eomes$^-$ cells, respectively (*Figure 8—figure supplement 1A*). With respect to our scRNA-seq analysis, these cells represent Eomes$^+$ (conventional) NK cells from uninfected and infected mice, and *T. gondii*-induced ILC1-like cells (both converted NK cells and Ly6C$^+$ ILC1s), of which most were converted NK cells (*Figures 6* and *7B*). Principal component analysis and hierarchical clustering showed a higher degree of similarity between NK cells from uninfected and infected mice, relative to ILC1-like cells (*Figure 8A,B*). As expected, *Ly6c1*, *Ly6c2*, *Itga1*, and *Eomes* were differentially expressed (*Figure 8C*). We found extensive differences between NK cells and ILC1-like cells, including differences in the expression of surface markers, transcription factors, secreted factors, adhesion molecules, chemokine receptors, and signaling molecules (*Figure 8D*).

Since NK cells and ILC1s display distinct patterns of chromatin accessibility (*Shih et al., 2016*), and infection can induce long-lasting epigenomic changes in NK cells (*Lau et al., 2018*), we performed the assay for accessibility for transposase-accessible chromatin using sequencing (ATAC-seq) (*Buenrostro et al., 2015*) to compare the NK cells and ILC1-like cells present in *T. gondii*-infected mice at the epigenomic level. We found widespread chromatin remodeling in ILC1-like cells, which clustered separately from NK cells (*Figure 8E,F*). Most differentially accessible regions fell within intronic and intergenic regions, consistent with categorization as putative enhancers (*Figure 8—figure supplement 1B*).

We organized differentially accessible regulatory elements (REs) into a non-redundant peak set containing six groups (*Figure 8—figure supplement 1C,D*). Groups 1, 3, and 5 contained REs that were uniquely more accessible in uninfected NK cells, NK cells from infected mice, and ILC1-like cells, respectively. Groups 2, 4, and 6 contained REs that were shared between two groups. Group 2 encompassed REs that were more accessible in both NK cell groups relative to ILC1-like cells, and accounted for the largest fraction of peaks, showing that NK cells from both uninfected and infected mice share many epigenetic features. Group 4, which contained REs that were more accessible in both NK cells and ILC1-like cells from infected mice relative to uninfected NK cells, was the second largest group, suggesting that infection may induce epigenetic changes common to both populations.

We directly compared the REs in groups 3 and 5 to identify additional differences between NK cells and ILC1-like cells following infection. Analysis of transcription factor binding motifs revealed motifs that were significantly enriched in each population, such as NF-κb in group 3, whose motif was present in 13.2% of REs, and Klf4 in group 5, whose motif was present in 25.8% of REs (*Figure 8—figure supplement 1E*). We also identified the specific loci that contained the most differentially accessible REs, and noted that the presence of more accessible REs generally correlated with greater gene expression. Notably, many of the epigenetic features shared by memory NK cells and memory CD8$^+$ T cells (*Lau et al., 2018*) were among the loci that gained accessibility in ILC1-like cells (*Figure 8—figure supplement 1F*). Among the differentially accessible regions were the *Eomes* and *Tbx21* loci, supporting our data above detailing expression changes of these transcription factors (*Figure 8G*, *Figure 8—figure supplement 1G*). Overall, our data suggest that *T. gondii* induces the conversion of NK cells into ILC1-like cells in a process that encompasses changes in gene expression and chromatin accessibility.

## Discussion

Infection with *T. gondii*, a natural mouse pathogen, causes the expansion of a novel ILC1-like subpopulation that stably persists even after acute infection resolves. These ILC1-like cells are heterogeneous, due to contributions from both ILC1s and NK cells, though NK cells give rise to most ILC1-like cells by means of Eomes downregulation. These findings disrupt current notions surrounding NK cells and ILC1s, which were recently established as discrete ILC lineages (*Vivier et al., 2018*), and

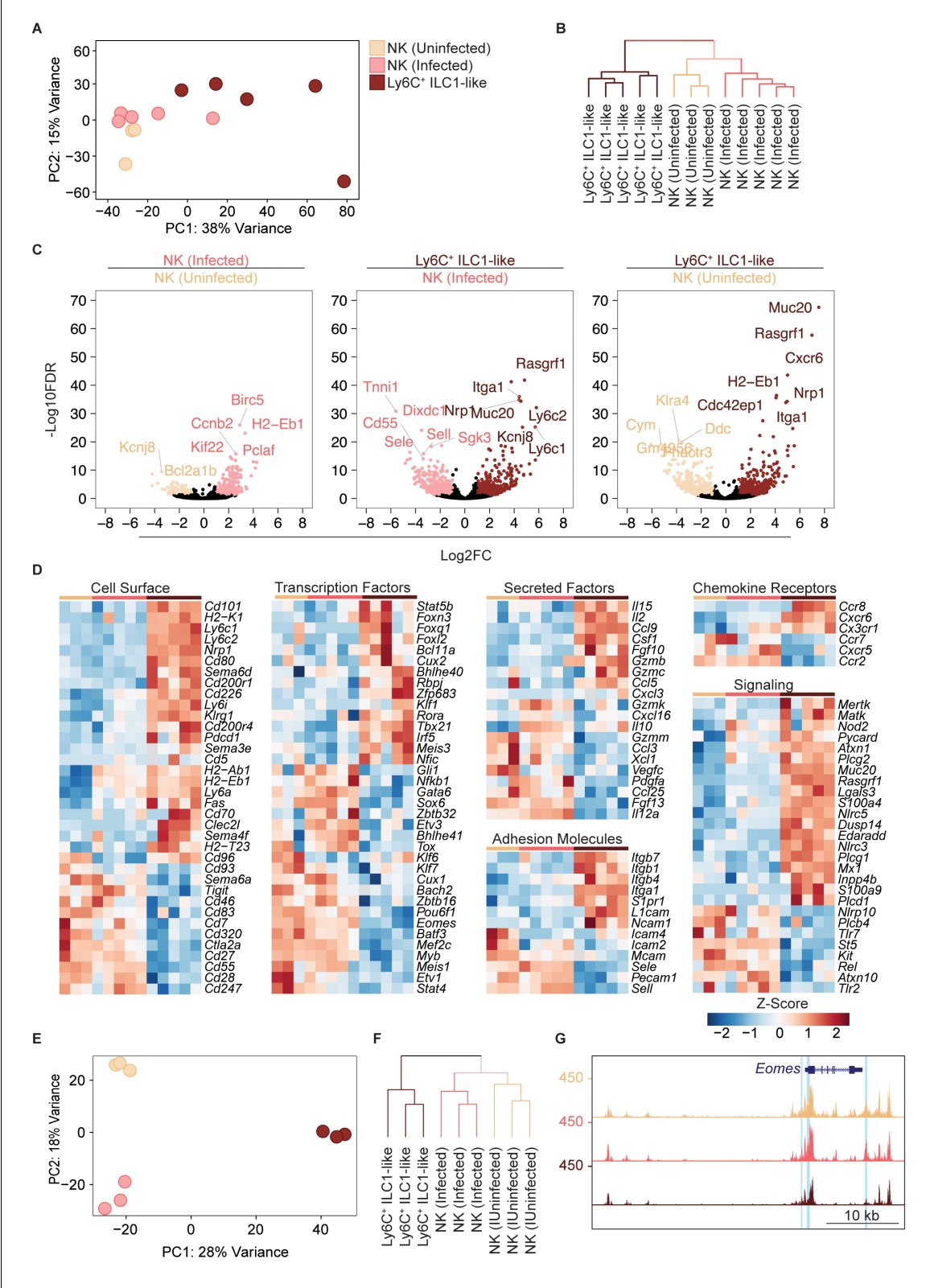

**Figure 8.** Eomes downregulation within NK cells accompanies extensive transcriptional and epigenomic changes. (A) Principal component analysis using top 2000 most variable genes from RNA-seq analysis comparing NK cells from spleens of uninfected and infected mice and Ly6C⁺ ILC1s (ILC1-like cells) from d35-infected mice, each dot represents a sample from different mice, *n* = 3 mice (NK cells-uninfected) or n = 5 mice (NK cells-infected and Ly6C⁺ ILC1-like). Color legend shown applies to all panels. (B) Unsupervised hierarchical clustering of each sample in (A). (C) Volcano plots showing

*Figure 8 continued on next page*

*Figure 8 continued*

Log₂(Fold Change) differences versus −Log₁₀(False Discovery Rate) for indicated comparisons, using pooled data from each indicated sample type. Genes exhibiting $Log_2FC > 1$ and $FDR < 0.1$ are colored. (D) Heatmaps showing centered DESeq2 variance-stabilized expression values of select genes exhibiting $Log_2FC > 1$ and $FDR < 0.1$, $n = 3$ mice (NK cells-uninfected) or $n = 5$ mice (NK cells-infected and Ly6C⁺ ILC1-like). Each column represents a different mouse sample of a given cell type, indicated by color code shown in (A). (E) Principal component analysis from ATAC-seq analysis comparing NK cells from spleens of uninfected and infected mice and Ly6C⁺ ILC1s (ILC1-like cells) from d35-infected mice, $n = 3$ mice. (F) Unsupervised hierarchical clustering of samples in (E). (G) Representative UCSC genome browser tracks showing ATAC-seq peaks in NK cells from uninfected mice, and NK cells and Ly6C⁺ ILC1s from d35-infected mice, at the *Eomes* locus. Differentially accessible REs are highlighted in blue.

DOI: https://doi.org/10.7554/eLife.47605.015

The following figure supplement is available for figure 8:

**Figure supplement 1.** ATAC-seq analysis reveals widespread changes in chromatin accessibility in *T*.

DOI: https://doi.org/10.7554/eLife.47605.016

exemplify the limitations of categorizing NK cells and ILC1s based solely on their phenotypes under steady-state conditions.

Previous studies indicated that Eomes downregulation may occur in NK cells within the tumor microenvironment (*Gill et al., 2012*; *Gao et al., 2017*) and in vitro (*Cortez et al., 2016*). Akin to the intermediate ILC1s (intILC1s) described within tumors (*Gao et al., 2017*), we found *T. gondii* infection induced Eomes⁺ CD49a⁺ NK cells in the liver, which are possibly intermediates between NK cells and ex-NK cells. However, our findings differ in several significant ways. The tumor studies indicated that Eomes⁻ NK cells were restricted to tumors and emphasized the importance of the microenvironment (*Gao et al., 2017*; *Gill et al., 2012*), concluding that within tumors, Eomes downregulation transiently silences NK cells to hamper immunosurveillance (*Silver and Humbles, 2017*). There was no evidence presented that these Eomes⁻ cells can circulate, though they proliferate when adoptively transferred into lymphocyte-deficient hosts. Moreover, it is not clear if these cells persist in the absence of tumor. By contrast, *T. gondii*-induced ILC1-like cells are disseminated throughout the circulation, retain the ability to produce IFNγ, do not produce TNFα, and are maintained in the absence of ongoing stimulus.

Our findings further indicate a permanent transformation of *T. gondii*-induced ILC1-like cells rather than a transient response to inflammation, a notion that is further bolstered by the observations that ILC1-like cells possess a unique gene expression profile. Moreover, *T. gondii* infection induced epigenetic modifications in NK cells and ILC1-like cells, although our analysis did not specifically examine each of the different individual NK and ILC1-like subsets we identified by scRNA-seq. However, these modifications were extensive and could not be readily explained by heterogeneity in the ILC1-like cells that we discovered by scRNA-seq. Nonetheless, our studies go beyond just the differences between tumor-induced and *T. gondii*-induced Eomes downregulation. Our findings indicate that DX5 and CD49a expression to identify NK cells and ILC1s may not always faithfully reflect Eomes expression, as is the case during early *T. gondii* infection. Moreover, although there is clear evidence supporting conversion from NK cells to Eomes⁻ NK cells, such as intILC1s (*Gao et al., 2017*), there is a general inability to distinguish between preexisting ILC1s and de novo converted ILC1s. Indeed, this may explain discrepancies that currently exist within the field, such as the conflicting report that ILC1-like cells confer protection against certain tumors (*Dadi et al., 2016*). Thus, our studies suggest that ILC1s may be more heterogeneous than currently thought, as they can arise from distinct precursors (*Vivier et al., 2018*).

In *T. gondii* infection, IL-12 is critical for NK cell activation (*Gazzinelli et al., 1993*) and our data suggest that it can have long-lasting effects. This parallels the requirements for development of memory and memory-like NK cells induced by MCMV and cytokine stimulation (*Cooper et al., 2009*; *Sun et al., 2012*; *Romee et al., 2012*). In addition to IL-12 dependence, MCMV-induced memory NK cells and *T. gondii*-induced ILC1-like cells share Ly6C expression, as well as alterations in their receptor repertoire, although ILC1-like cells primarily express the inhibitory receptor NKG2A while MCMV-induced memory NK cells are Ly49H⁺. Moreover, memory and memory-like NK cells have few markers that differentiate them from resting NK cells, whereas NK cells converted by *T. gondii* infection lose expression of Eomes and gain expression of Ly6C, KLRG1, CX₃CR1, and Nrp-1,

with scRNA-seq revealing global transcriptional differences that identify them as a discrete subpopulation.

The persistence of ILC1-like cells after clearance of infection is reminiscent of classical immune memory and suggests that ex-NK cells play a role beyond what has been attributed to NK cells and ILC1s in *T. gondii* infection thus far. One possibility is that ILC1-like cells are an inflammatory sub-population, such as the inflammatory ILC2s that gain circulatory capacity in response to activation by cytokines or infection (*Huang et al., 2018*). These inflammatory ILC2s can be distinguished from tissue-resident ILC2s based on their increased expression of KLRG1 and S1P receptors, features that are also characteristic of *T. gondii*-induced ILC1-like cells, but not steady-state tissue-resident NK cells. Most ILC1-like cells express $CX_3CR1$, which has previously been shown to regulate NK cell recruitment into the circulation (*Ponzetta et al., 2013*; *Sciume et al., 2011*). Another possibility is that ILC1-like cells may provide a protective role in subsequent *T. gondii* infections, as shown previously (*Denkers et al., 1993*), although this finding remains controversial (*Goldszmid et al., 2007*). Moreover, we have not been able to show that repeated injection of an avirulent *T. gondii* strain confers NK1.1-dependent protection against subsequent lethal re-challenge. Regardless, we have observed ILC1-like cell expansion following repeated exposure to avirulent *T. gondii* strains, and future studies may illuminate how gaining circulatory capacity affects the functionality of ILC1-like cells, and how these cells may contribute to subsequent immune responses to *T. gondii*.

In summary, we found that *T. gondii* infection induced permanent changes to NK cells and ILC1s, including conversion of NK cells into ILC1-like cells. Our studies indicate that the current system of ILC classification, based on their phenotype and development under steady state conditions, may not apply following inflammation. Rather, plasticity may give rise to populations that resemble one another at first glance, but actually represent the convergence of multiple developmental paths to form interrelated populations.

# Materials and methods

## Key resources table

| Reagent type (species) or resource | Designation | Source or reference | Identifiers | Additional information |
|---|---|---|---|---|
| Chemical compound, drug | D-Luciferin, Potassium Salt | Gold Biotechnology | Cat#LUCK-1G; CAS 115144-35-9 | |
| Chemical compound, drug | Sulfadiazine | Sigma | Cat#S8626-25G | |
| Chemical compound, drug | Uracil | Sigma | Cat#U1128 | |
| Chemical compound, drug | Percoll | Sigma | Cat#P1644 | |
| Chemical compound, drug | Recombinant murine IL-12 | Pepro Tech | Cat#210–12 | |
| Chemical compound, drug | Recombinant mouse IL-18 | MBL | Cat#B002-5 | |
| Chemical compound, drug | Recombinant murine IL-15 | Peptro Tech | Cat#210–15 | |
| Chemical compound, drug | DNase I | Sigma | Cat#10104159001 | |
| Chemical compound, drug | Collagenase Type IV | Sigma | Cat#C5138-100mg | |
| Commercial assay or kit | Foxp3/Transcription Factor Staining Buffer Set | eBioscience | Cat#00-5523-00 | |
| Commercial assay or kit | EasySep Magnet | BD Biosciences | Cat#1800 | |

*Continued on next page*

*Continued*

| Reagent type (species) or resource | Designation | Source or reference | Identifiers | Additional information |
|---|---|---|---|---|
| Commercial assay or kit | EasySep Mouse Streptavidin Rapidspheres Isolation Kit | BD Biosciences | Cat#19860 | |
| Commercial assay or kit | Nextera Index Kit | Illumina | Cat#FC-121–1011 | |
| Commercial assay or kit | SMARTer PCR cDNA Synthesis Kit | Clontech | Cat#634926 | |
| Genetic reagent (*Homo sapiens*) | Human: HFF | John C. Boothroyd | | |
| Cell line (*Rattus norvegicus*, *Mus musculus*) | 2.4G2 hybridoma | ATCC | ATCC HB-197 | |
| Cell line (*Rattus norvegicus*, *Mus musculus*) | PK136 hybridoma | ATCC | ATCC HB-191 | |
| Genetic reagent (*Mus musculus*) | wild-type; C57BL/6NCr | Charles River | Cat#556 | |
| Genetic reagent (*Mus musculus*) | CD45.1$^{+;}$ B6.SJL-*Ptprc$^a$Pepc$^b$*/BoyCrCrl | Charles River | Cat#564 | |
| Genetic reagent (*Mus musculus*) | Balb/c; BALB/cAnNCr | Charles River | Cat#555 | |
| Genetic reagent (*Mus musculus*) | NKp46$^{iCre}$; Ncr1$^{tm1.1(icre)Viv}$/J | Eric Vivier | MGI:5308410 | |
| Genetic reagent (*Mus musculus*) | *Eomes$^{f/f}$*; B6.129S1(Cg)-Eomes$^{tm1.1Bflu}$/J | The Jackson Laboratory | MGI:4830338, Cat#017293 | |
| Genetic reagent (*Mus musculus*) | *Stat4$^{-/-}$*; C.129S2-Stat4$^{tm1Gru}$/J | The Jackson Laboratory | MGI: 1857248, Cat#002826 | |
| Genetic reagent (*Mus musculus*) | *Tbx21$^{-/-}$*; B129. S6-Tbx21$^{tm1Glm}$/J | The Jackson Laboratory | MGI: 2180194, Cat#004648 | |
| Genetic reagent (*Mus musculus*) | *Tgfbr2$^{f/f}$*; B6;129-Tgfbr2$^{2tm1Karl}$/J | The Jackson Laboratory | Cat#012603, MGI: 98729 | |
| Genetic reagent (*Mus musculus*) | *Zeb2*-GFP; *Zfhxlb$^{tm2.1Yhi}$* | Kenneth Murphy | | |
| Genetic reagent (*Toxoplasma gondii*) | Prugniaud; Pru; PRU-FLuc-GFP | John C. Boothroyd | | |
| Genetic reagent (*Toxoplasma gondii*) | WT *T. gondii*; PruΔku80 Δhx | L. David Sibley | | |
| Genetic reagent (*Toxoplasma gondii*) | Δgra4 *T. gondii*; PruΔku80 Δhx Δgra4::HX/mcherry | L David Sibley | | |
| Software, algorithm | Flowjo 10 | Treestar | https://www.flowjo.com/ ; RRID: SCR_008520 | |
| Software, algorithm | Prism 7 | Graphpad | https://www.graphpad.com/ ; RRID: SCR_002798 | |
| Software, algorithm | Bowtie2 version 2.4.3.2b | *Langmead and Salzberg, 2012* | http://bowtie-bio.sourceforge.net/bowtie2/manual.shtml | |
| Software, algorithm | Samtools version 1.4 | *Li et al., 2009* | http://samtools.sourceforge.net/ | |
| Software, algorithm | Picard Tools version 2.18.14 | Broad Institute | http://broadinstitute.github.io/picard/ | |
| Software, algorithm | BEDTools version 2.27.0 | *Quinlan and Hall, 2010* | https://bedtools.readthedocs.io/en/latest/ | |

*Continued on next page*

*Continued*

| Reagent type (species) or resource | Designation | Source or reference | Identifiers | Additional information |
|---|---|---|---|---|
| Software, algorithm | MACS2 version 2.1.0 | *Zhang et al., 2008* | https://github.com/taoliu/MACS | |
| Software, algorithm | deepTools version 3.1.2 | *Ramírez et al., 2014* | https://deeptools.readthedocs.io/en/develop/ | |
| Software, algorithm | Homer version 4.10 | *Heinz et al., 2010* | http://homer.ucsd.edu/homer/ | |
| Software, algorithm | RStudio version 3.5.1 | R Core Team 2014 | https://www.r-project.org/ | |
| Software, algorithm | Bioconductor package DESeq2 version 1.21.22 | *Love et al., 2014* | http://bioconductor.org/packages/release/bioc/html/DESeq2.html | |
| Software, algorithm | Kallisto version 0.44.0 | *Bray et al., 2016* | https://pachterlab.github.io/kallisto/download | |
| Software, algorithm | GSEA | *Mootha et al., 2003*; *Subramanian et al., 2005* | http://software.broadinstitute.org/gsea/index.jsp | |
| Software, algorithm | Seurat | *Butler et al., 2018* | https://cran.r-project.org/web/packages/Seurat/index.html | |

## Mice

*Eomes^{f/f}*, *Tbx21^{-/-}*, and *Stat4^{-/-}* were purchased from The Jackson Laboratory. C57BL/6, CD45.1[+], and BALB/c mice were purchased from Charles River. *Ncr1^{iCre}* mice were a kind gift from Eric Vivier. *Zeb2*-GFP mice were a generous gift from Kenneth Murphy (*Wu et al., 2016*). Eomes-GFP mice were graciously provided by Thierry Walzer (*Daussy et al., 2014*). *Ncr1^{iCre}* were bred to *Eomes^{f/f}* mice to generate *Eomes* cKO mice and Cre-negative littermate controls. *Rag2^{-/-}* were bred to *Il2rg^{-/-}* to generate *Rag2^{-/-} Il2 rg^{-/-}* mice. Age and sex-matched animals were used in all experiments. Mice were infected at 6–12 wk of age. All protocols were approved by the Institutional Animal Care and Uses Committee (Washington University School of Medicine, St. Louis, MO) under animal protocol number 20160002.

## Parasites

PRU-FLuc-GFP Type II strain of *T. gondii* was a generous gift from John C. Boothroyd and used in all infections unless otherwise specified. PruΔku80Δhxδgra4::HX/mCherry and the wildtype parental strain PruΔku80Δhx were previously described (*Jones et al., 2017*). The attenuated *Cps1-1* mutant on the type I RH background was described previously (*Fox and Bzik, 2002*).

All *T. gondii* strains used in this study were maintained in human foreskin fibroblast (HFF) monolayers grown in D-10 medium (Dulbecco's modified Eagle medium, 10% fetal bovine serum (FBS), 2 mM glutamine, 10 mM HEPES pH 7.5, 20 µg/mL gentamicin), maintained at 37°C with 5% $CO_2$. The *Cps1-1* mutant was grown as described above but supplemented with 200 µM uracil (Sigma). Mature parasites were lysed from host cells by vigorous pipetting and egressed parasites were filtered through 3 µm polycarbonate membranes and resuspended in HHE medium (Hanks' balanced salt solution, 10 mM HEPES, 0.1 mM EGTA). Cell cultures were determined to be mycoplasma-negative using the e-Myco plus kit (Intron Biotechnology).

## Infection

Mice were injected intraperitoneally (i.p.) with 200 PRU-FLuc-GFP, PruΔku80Δhx, or PruΔku80Δhxδgra4::HX/mCherry tachyzoites that were propagated in culture as described above. In indicated experiments, mice were treated with 0.5 g/L sulfadiazine in drinking water to suppress parasite growth. To generate non-persistent infections, $1 \times 10^5$ *Cps1-1* tachyzoites were injected three times, 2 weeks apart.

## Luciferase imaging

Mice were injected i.p. with 150 mg D-luciferin (Gold Biotechnology) per kg body weight, incubated for 10 min, then anesthetized with continuous isoflurane anesthesia at a flow rate of 1 L/min. Images were captured using an IVIS Spectrum In Vivo Imaging System (Perkin Elmer). Luminescence was quantitated using Living Image software (Perkin Elmer). NK1.1-depleted mice and control mice were injected i.p. with 100 µg of anti-NK1.1 or isotype control antibody 3 days and 1 day prior to infection. Purified anti-NK1.1 antibody and isotype control were generated at the Rheumatic Diseases Core Center Protein Purification and Production Facility using the PK136 hybridoma (ATCC) and MAR 18.5 hybridoma (ATCC), respectively.

## Cell isolation

Spleens were mashed through a 70 µm cell strainer and treated with Tris-NH$_4$Cl to lyse red blood cells. Livers were mashed through a 70 µm cell strainer, resuspended in isotonic 38.5% Percoll (Sigma- Aldrich), centrifugated at 325 x g for 20 min, and treated with Tris-NH$_4$Cl. Peritoneal cells were isolated by lavage of the peritoneal cavity with PBS. Bone marrow was flushed from femurs and tibias, mashed through a 70 µm cell strainer, and treated with Tris-NH$_4$Cl. Lymph nodes were mashed through a 70 µm cell strainer. Brains were mashed through a 70 µm cell strainer, resuspended in 38.5% isotonic Percoll, and centrifugated at 325 x g for 20 min. Lungs were perfused with PBS, minced, digested in RPMI-1640 containing 2% FBS, 1 mg/mL Collagenase Type IV (Sigma-Aldrich), and 0.2 mg/mL DNase (Sigma-Aldrich), mashed through a 70 µm cell strainer, resuspended in 38.5% isotonic Percoll, and centrifugated at 325 x g for 20 min. Uterus and salivary gland were minced, digested in RPMI-1640 containing 0.17 mg/mL Liberase TL (Sigma-Aldrich) and 0.1 mg/mL DNase (Roche), mashed through a 70 µm cell strainer, resuspended in 38.5% isotonic Percoll, and centrifugated at 325 x g for 20 min.

## Flow cytometry and cell sorting

Cells were stained with Fixable Viability Dye and F$_c$ receptors were blocked with 2.4G2 hybridoma (ATCC) culture supernatants prior to staining in PBS containing 2% FBS and 2.5 mM EDTA on ice. Data was acquired using a FACSCanto instrument (BD Biosciences) or FACSAria instrument (BD Biosciences) using FACSDiva software (BD Biosciences). Data was analyzed with Flowjo v10 (Treestar).

For cell sorting, splenocytes were enriched for NK1.1$^+$ NKp46$^+$ cells by negative selection. Splenocytes were incubated with 0.2 µg/mL biotin mouse CD4, 0.2 µg/mL biotin anti-mouse CD8, and 1 µg/mL biotin anti-mouse CD19. One hundred µL of EasySep Mouse Streptavidin Rapidspheres (BD Biosciences) were added per $1 \times 10^8$ splenocytes. Cells were placed in an EasySep magnet (BD Biosciences) and enriched cells were poured off and subjected to extracellular staining. Sorting was performed using a FACSAria instrument (BD Biosciences) using FACSDiva software (BD Biosciences). NK cells were identified as CD3$^-$ CD19$^-$ NK1.1$^+$ NKp46$^+$ Eomes-GFP$^+$ CD49a$^-$ and *T. gondii*-induced ILC1-like cells were identified as CD3$^-$ CD19$^-$ NK1.1$^+$ NKp46$^+$ Eomes-GFP$^-$ CD49a$^+$.

## Stimulations and intracellular staining

Following extracellular staining, cells were fixed with FoxP3/Transcription Factor Staining Buffer Set for 30 min at room temperature and washed with 1x Permeabilization buffer (eBiosciences). Transcription factor antibodies were diluted in 1x FoxP3 permeabilization buffer and cells were incubated for 30 min at room temperature.

To assess Eomes downregulation after culture in IL-12, $5 \times 10^6$ splenocytes were cultured in R-10 medium (RPMI-1640 medium containing 10% FBS, 2 mM glutamine, 100 U/mL penicillin, and 100 µg/mL streptomycin) with 300 IU/mL IL-2 with 20 ng/mL IL-12 (Pepro Tech). Unstimulated control wells contained only 300 IU/mL IL-2.

For detection of IFNγ production, $5 \times 10^6$ splenocytes were cultured for 5 hr in R-10, with the addition of 20 ng/mL IL-12, 20 ng/mL IL-12 + 5 ng/mL IL-18, or 0.5 µg/mL PMA + 4 µg/mL Ionomycin. Brefeldin A was added after 1 hr.

## Generation of bone marrow chimeras

Donor bone marrow was harvested by flushing femurs and tibias and mashing through a 70 µm cell strainer. Cells were mixed in a 1:1 ratio with CD45.1$^+$ bone marrow, and intravenously injected into

irradiated mice. Recipient mice were lethally irradiated (900 rad) and reconstituted with a 1:1 ratio of CD45.1$^+$ bone marrow cells and CD45.2$^+$ *Eomes* cKO, *Eomes*$^{f/f}$, *Tbx21*$^{-/-}$, or *Stat4*$^{-/-}$ bone marrow cells (*Kaplan et al., 1996*; *Zhu et al., 2010*; *Finotto et al., 2002*). Mice were maintained on sulfamethoxazole/trimethoprim oral suspension added to the drinking water for 2 weeks after reconstitution and used for experiments 6–8 weeks after reconstitution. To prevent chimeric mice from succumbing to acute *T. gondii* infection, bone marrow chimeras were treated with 0.5 g/L sulfadiazine in drinking water beginning 10 d.p.i. and maintained on sulfadiazine until the end of the experiment.

## Single-cell RNA-Seq

Sorted cells were subjected to droplet-based 3' end massively parallel single-cell RNA sequencing using Chromium Single Cell 3' Reagent Kits as per manufacturer's instructions (10x Genomics). The libraries were sequenced using a HiSeq3000 instrument (Illumina).

Sample demultiplexing, barcode processing, and single-cell 3' counting was performed using the Cell Ranger Single-Cell Software Suite (10x Genomics). Cellranger count was used to align samples to the mm10 reference genome, quantify reads, and filter reads with a quality score below 30.

The Seurat package in R was used for subsequent analysis (*Butler et al., 2018*). Genes expressed in fewer than 3 cells and cells that expressed less than 400 or greater than 3500 genes were removed for downstream analysis. Data was normalized using a scaling factor of 10,000 s and nUMI was regressed with a negative binomial model. Principal component analysis was performed using the top 3000 most variable genes and t-SNE analysis was performed with the top 40 PCAs. Clustering was performed using a resolution of 0.8. For heatmaps, the mean expression by all cells within the cluster was used.

## RNA sequencing

RNA was isolated from $5 \times 10^4$ sorted cells using Trizol. Libraries were prepared using the Clontech SMARTer Kit. Sequencing was performed using $1 \times 50$ single-end reads with a HiSeq3000 instrument (Illumina). Reads were quantified using kallisto and differential expression was assessed using the DESeq2 package in R (*Bray et al., 2016*; *Love et al., 2014*). Using a log2FC cutoff of 1 and an FDR threshold of 0.1, we identified 657 differentially expressed (DE) genes between the NK cells from uninfected and d35-infected mice, 2288 DE genes between NK cells from infected mice and *T. gondii*-induced ILC1-like cells, and 1685 DE genes between NK cells from uninfected mice and *T. gondii*-induced ILC1-like cells. Variance-stabilized transform values were used for subsequent analysis. All plots were generated in R using the ggplot2, pheatmap, and hclust packages. GSEA was performed using the GSEA Preranked module in GenePattern (*Subramanian et al., 2005*; *Mootha et al., 2003*; *Reich et al., 2006*).

## ATAC-seq

Samples were prepared as previously described (*Buenrostro et al., 2015*), then purified with MinElute spin columns (Qiagen). DNA fragments were amplified using Nextera index adapters as per manufacturer's instructions (Illumina). Libraries were purified with AMPure XP beads (Beckman Coulter). Three libraries were pooled and sequenced with $2 \times 50$ paired end reads using a HiSeq3000 instrument (Illumina).

Sequences were aligned to the mm10 reference genome using Bowtie2 (*Langmead and Salzberg, 2012*). Reads with a quality score below 30 were removed with Samtools (*Li et al., 2009*) and duplicate reads were filtered with PicardTools (Broad Institute). Peaks were called using MACS2 (*Zhang et al., 2008*) with an FDR cutoff of 0.05. Narrowpeak files generated from MACS2 were converted to Bigwig files with deepTools (*Ramírez et al., 2014*), and visualized using the UCSC genome browser. Differential peaks were identified using Homer (*Heinz et al., 2010*) with a Log2FC > 1 and an FDR cutoff of 0.05. Heatmaps of differentially accessible regions were generated with deepTools. Transcription factor binding motifs were identified de novo with Homer.

Across all samples, we identified 71,504 peaks. We further analyzed 9640 discrete peaks that displayed two-fold changes between at least two groups. Between NK cells from control and infected mice, there were 1809 differential peaks (1445 were larger in the NK cells from infected mice and 364 were larger in NK cells from uninfected mice), between NK cells from infected mice and *T.*

*gondii*-induced ILC1-like cells, there were 6710 differential peaks (4572 were larger in the NK cells and 2138 were larger in *T. gondii*-induced ILC1-like cells), and between NK cells from uninfected mice and *T. gondii*-induced ILC1-like cells, there were 5938 differential peaks (3065 were larger in the uninfected NK cells and 2973 were larger in *T. gondii*-induced ILC1-like cells).

### Statistical analysis

Prism (GraphPad) was used for statistical analysis. Student's t-test was used for comparison of two groups and two-way ANOVA with Bonferroni correction was used in analyses involving multiple comparisons. p-Values are shown in figures and figure legends. In all graphs, points represent biological replicates, bar position represents the mean, and error bars represent + s.e.m.

## Acknowledgements

This work was supported by NIH grants F30DK108472 (EP), AI128845 (WMY), AI120606 (EMO), AI134035 (EMO, MC), AI11852 (EMO), and CA188286 (EMO). We thank members of the Yokoyama Lab, Ilija Brizic, Stipan Jonjic, and Victor Cortez for helpful discussions, Jennifer Barks and Reeha Savari Dhason for HFF cells, Nathaniel Jones, Sumit Kumar, Josh Radke, and Kevin Brown for *T. gondii* strains, Suzanne Hickerson for technical assistance, and Lacey Feigl, Lorraine Schwartz, and Shirley McTigue for administrative assistance.

We thank the Genome Technology Access Center in the Department of Genetics at Washington University School of Medicine for help with genomic analysis. The Center is partially supported by NCI Cancer Center Support Grant #P30 CA91842 to the Siteman Cancer Center and by ICTS/CTSA Grant# UL1 TR000448 from the National Center for Research Resources (NCRR), a component of the National Institutes of Health (NIH), and NIH Roadmap for Medical Research. This publication is solely the responsibility of the authors and does not necessarily represent the official view of NCRR or NIH.

We also thank the McDonnell Genome Institute, especially Catrina Fronick, Robert Fulton, and Amy Ly, for assistance and advisement regarding genomic analysis.

## Additional information

### Funding

| Funder | Grant reference number | Author |
| --- | --- | --- |
| National Institute of Diabetes and Digestive and Kidney Diseases | F30DK108472 | Eugene Park |
| National Institute of Allergy and Infectious Diseases | AI120606 | Eugene Oltz |
| National Institute of Allergy and Infectious Diseases | AI134035 | Marco Colonna Eugene Oltz |
| National Institute of Allergy and Infectious Diseases | AI11852 | Eugene Oltz |
| National Cancer Institute | CA188286 | Eugene Oltz |
| National Institute of Allergy and Infectious Diseases | AI128845 | Wayne M Yokoyama |

The funders had no role in study design, data collection and interpretation, or the decision to submit the work for publication.

### Author contributions

Eugene Park, Conceptualization, Resources, Data curation, Formal analysis, Funding acquisition, Validation, Investigation, Visualization, Methodology, Writing—original draft, Writing—review and editing; Swapneel Patel, Investigation, Writing—review and editing; Qiuling Wang, Beatrice Plougastel-Douglas, Resources, Methodology; Prabhakar Andhey, Data curation, Formal analysis, Visualization; Konstantin Zaitsev, Patrick Collins, Data curation, Formal analysis; Sophia Porter,

Resources, Investigation; Maxwell Hershey, Michael Bern, Validation, Investigation; Marco Colonna, Conceptualization, Resources, Funding acquisition; Kenneth M Murphy, Resources, Supervision; Eugene Oltz, Maxim Artyomov, L David Sibley, Resources, Supervision, Methodology; Wayne M Yokoyama, Conceptualization, Formal analysis, Supervision, Funding acquisition, Visualization, Writing—original draft, Project administration, Writing—review and editing

### Author ORCIDs

Eugene Park (iD) http://orcid.org/0000-0002-2617-7571
Marco Colonna (iD) http://orcid.org/0000-0001-5222-4987
Wayne M Yokoyama (iD) https://orcid.org/0000-0002-0566-7264

### Ethics

Animal experimentation: All protocols were approved by the Institutional Animal Care and Uses Committee (Washington University School of Medicine, St. Louis, MO) under animal protocol number 20160002.

### Decision letter and Author response

Decision letter https://doi.org/10.7554/eLife.47605.023
Author response https://doi.org/10.7554/eLife.47605.024

## Additional files

### Supplementary files

• Transparent reporting form
DOI: https://doi.org/10.7554/eLife.47605.017

### Data availability

RNA-seq and ATAC-seq data are available on Gene Expression Omnibus under the accession number GSE124313. scRNA-seq data are available under accession number GSE124577.

The following datasets were generated:

| Author(s) | Year | Dataset title | Dataset URL | Database and Identifier |
|---|---|---|---|---|
| Park E | 2019 | Gene transcription and chromatin accessibility of conventional NK cells and ILC1s in spleens of uninfected and Toxoplasma gondii-infected Mice | https://www.ncbi.nlm.nih.gov/geo/query/acc.cgi?acc=GSE124313 | NCBI Gene Expression Omnibus, GSE124313 |
| Park E | 2019 | Toxoplasma gondii Infection Promotes NK Cell Conversion into ILC1s and Heterogeneous ILC1 Populations | https://www.ncbi.nlm.nih.gov/geo/query/acc.cgi?acc=GSE124577 | NCBI Gene Expression Omnibus, GSE124577 |

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
