## [Decision Letter]

Thank you for submitting your article "*Toxoplasma gondii* infection drives conversion of NK Cells into ILC1s" for consideration by *eLife*. Your article has been reviewed by Tadatsugu Taniguchi as the Senior Editor, a Reviewing Editor, and three reviewers. The following individuals involved in review of your submission have agreed to reveal their identity: Chris Hunter (Reviewer #1); Jinfang Zhu (Reviewer #2); Eric O Long (Reviewer #3).

The reviewers have discussed the reviews with one another and the Reviewing Editor has drafted this decision to help you prepare a revised submission.

Summary:

Natural Killer (NK) and the ILC1 subset of innate lymphoid cells share related functions in host defense but have been argued to arise from distinct pathways. Park et al. present new evidence challenging this concept. They show that murine *Toxoplasma gondii* infection promotes the differentiation of NK cells into an ILC1-like cell population which is stable and long-lasting, even after the infection has been cleared. These *T.gondii* induced cells, unlike Eomes^+^CD49a^-^ NK cells, are Eomes-CD49a^+^T-bet+ and therefore resemble ILC1 cells. The authors additionally show that their differentiation involves Eomes down regulation and is STAT-4 dependent, However, in common with NK cells and distinct from ILC1 the *T. gondii* induced "ILC-like" population circulates to blood and lungs. Finally, the authors employ single cell RNA-seq to examine the heterogeneity of the major *T. gondii* induced innate lymphocyte populations and their NK vs ILC relatedness as assessed by gene expression. Together, their observations establish a previously unappreciated developmental link between NK and ILC1 cells in the context of infection.

The three reviewers and editor agree that this is an important contribution that sheds new light on the developmental relationship of NK and ILC1 cells, a scientific issue that has received considerable attention in the innate immunity field. Although extensive, most of the criticisms raised can be addressed by revisions to the manuscript. One additional experiment is requested to provide a missing control.

Essential revisions:

All reviewers had a major concern about how this new population of *T. gondii* induced innate cells should be referred to in the manuscript. Based on the single cell RNA-seq data, these cells (cluster 10) are still closer to NK cells than to ILC1s (Figure 5F and Figure 6—figure supplement 1E) despite their loss in Eomes expression and acquisition of CD49a expression. Thus, one could easily think of them as "Eomes negative NK" or "ex-NK" cells rather than ILC1s, and to simply refer to them as Eomes-CD49a^+^ ILC1 cells may be misleading. For this reason, the authors should modify the title of the paper and change their designation throughout the manuscript. We suggest "ILC1-like" as a good descriptor. In addition, although it is clear that the "Eomes negative NK" cells that are generated during *T. gondii* infection are transcriptionally and epigenetically distinct from the NK cells in the steady state and NK cells after infection (Figure 7 and Figure 8—figure supplement 1), these "Eomes negative NK" cells referred to as "*T. gondii*-induced ILC1s" were not directly compared with classical ILC1s. Based on the single cell RNA-seq data, these cells may not express many of the ILC1-related signature genes. Therefore, again, the authors need to be cautious in referring to them as ILC1 cells.

A second concern was that the NK 1.1 depletion shown in Figure 1—figure supplement 1 was performed with a PBS rather than isotope matched immunoglobulin control which is considered unacceptable. The authors should repeat at least once with proper control Ig to make sure this is not issue. It is not necessary to repeat entire survival curve just experiments shown in A and B and initial survival to make sure there is no death in controls vs. antibody treated.

1) Figure 3E-H is focused on the liver, where populations of memory and tissue-resident NK cells reside. Do the authors have data sets for the liver in Figure 3A-C? Do they express Ly6C? Including them in the bar graph of Figure 3C would be useful to identify if the liver trends with the lungs or with the brain (both sites of parasite infection/replication)…or alternatively authors should continue to focus on the lung throughout the entirety of Figure 3.

2) The conclusion of Figure 3F is that "Eomes expression generally remained the same". However, for the representative flow plot of liver NK cells, the authors chose the lowest% of Eomes negative, while in the bar graph the rest of the points show that 20-50% of Eomes^+^ NK cells became Eomes-ve after transfer and migration to the liver. Please reword statement appropriately as Eomes expression is really not the same when the data are pooled.

3) The interesting point of Figure 3G is the upregulation of CD49a on Eomes^+^ NK cells post-infection. However, this population is eliminated from the subsequent histograms showing Ly6C expression. It may be of interest to note whether Ly6C expression by Eomes^+^CD49a^+^ cells is more similar to ILC1s or NK cells, or occurs at an intermediate level (which would make sense if that population is being seeded from both NK cells and ILC1s).

4) There are a lot of data represented in Figure 4C-D, without much rationale or interpretation provided in the text suggesting that items may be more appropriate for a supplemental figure.

5) The CD45.2/.1 appears to be switched on Figure 6—figure supplement 1A.

6) Although the contour plots don't show outliers, the bottom panel of Figure 3E says 3.66%, but there's nothing in the gate, which is a bit misleading.

7) The% of Eomes cKO during infection is missing in Figure 6D (bottom).

8) The marker Ly6C figures prominently in most of the paper. What is the significance of its expression and of its induction by *T. gondii* infection? Please provide background.

9) A recurrent problem is insufficient description of Figures. In several cases (e.g. Figure 4, Figure 5A), the source of cells used for analysis is not given in the Figure, the legend or the text. It would help to include the source of cells, eg liver or spleen, in each of these places.

10) As suggested above it would strengthen the manuscript if the authors should characterize the Eomes^+^CD49a^+^ cells found in the liver after *T. gondii* infection in a greater detail. Do they also express Ly6C?

11) In Figure 3F, ~30% of "ILC1s" re-express Eomes upon transfer; this should be discussed.

12) Figure 3D, the cell labels appear to be switched. Please check.

13) IL-7Ris an important marker to distinguish NK cells from ILC1s but the cells in cluster 10 do not express IL-7R as shown in Figure 5F. How about IL-2Rb? If possible the dependence of these NK/ILC1 subsets on IL-7 and/or IL-15 should be either tested or discussed.

14) Figure 3G is confusing. While there are 3 subsets for the combination of CD49a and Eomes expression (upper flow plots) only 2 are shown on the lower panels for Ly6C expression. What are they? Are the colors supposed to match between the top and low panels? Even so, it is not clearly visible.

15) Subsection “Expansion of ILC1s among circulating cells”. "Ly6C was expressed by a higher percentage." Percentage of what?

16) A number of different markers were used to distinguish among cell subsets, as for example in Figure 4. In many cases, no reason is given for the choice of markers. On what basis were they selected? Besides a phenotypic distinction of subsets, do they contribute to an understanding of the function of these different cells? Please clarify in each case.

17) In Figure 4D, lower panels, p values are indicated for different comparisons between markers. The main comparisons should be between uninfected and infected cells, and between NK and ILC1. For what reason are NK cells from uninfected mice compared to ILC from infected mice for every marker measured?

18) Figure 5G, the color gradient, used to presumably show expression level or frequency, does not allow visual discrimination. A different color scale is needed. Some scales go up to 2, others up to 5. What do they indicate?

19) Subsection “NK Cells downregulate Eomes during *T. gondii* infection”, "the numbers of CD49a^+^ and CD49a^+^ Ly6C^+^ cells were significantly reduced in Eomes cKO compared to Eomes f/f controls (Figure 6A,B)." That result is in 6B. Figure 6A is very different, it shows that the frequency of these cells is higher in Eomes cKO. This difference is not explained. In contrast, the experiment with bone marrow chimeras shown in Figure 6D is very clear. Why are changes in the relative cell frequencies so different from changes in cell numbers?

20) The columns in Figure 7D are not defined. The corresponding legend is dense. It looks like columns may have been grouped as NK-uninf, NK-inf and ILC1. Does each column represent a different mouse?

21) Subsection “Eomes downregulation within NK cells accompanies extensive transcriptional and epigenomic changes”. "NK cells share many epigenetic features…" Share with whom?

22) Subsection “Eomes downregulation within NK cells accompanies extensive transcriptional and epigenomic changes”. "Somewhat surprisingly" Since *T. gondii* infection has a large impact on both NK cells and ILC1 cells, why are epigenetic changes common to both populations surprising?

23) Sequencing analysis shows there are two genes for Ly6C. Does the analysis of phenotypes with anti-Ly6C antibody detect both? What is the significance of having two genes?

24) Subsection “*T. gondii* infection results in ILC1 expansion.”. A symbol that is neither an "a" nor a Greek letter α appears in "a-NK1.1". What is it? The context suggests it is an antibody that causes depletion of NK1.1^+^ cells in mice. If so, it should be described. What does the symbol represent? The symbol appeared again on subsection “*T. gondii*-induced ILC1s are distinct from NK Cells and ILC1s” in "TNFa". It should be replaced by the Greek letter α.

25) Visually, Figure 8—figure supplement 1C does not convey a clear message. How were the subsets defined? What could be their biological relevance?

26) The two comparisons made from the ATAC-seq analysis in Figure 7G for Eomes and Figure 8—figure supplement 1F for T-bet show small differences. Is there a way to express a statistical significance? The changes do not look convincing.

27) The legend to Figure 8—figure supplement 1E Figure does not adequately explain the Figure. What does the figure show?

---

## [Author Response]

Essential revisions:All reviewers had a major concern about how this new population of T. gondii induced innate cells should be referred to in the manuscript. Based on the single cell RNA-seq data, these cells (cluster 10) are still closer to NK cells than to ILC1s (Figure 5F and Figure 6—figure supplement 1E) despite their loss in Eomes expression and acquisition of CD49a expression. Thus, one could easily think of them as "Eomes negative NK" or "ex-NK" cells rather than ILC1s, and to simply refer to them as Eomes-CD49a^+^ ILC1 cells may be misleading. For this reason, the authors should modify the title of the paper and change their designation throughout the manuscript. We suggest "ILC1-like" as a good descriptor. In addition, although it is clear that the "Eomes negative NK" cells that are generated during T. gondii infection are transcriptionally and epigenetically distinct from the NK cells in the steady state and NK cells after infection (Figure 7 and Figure 8—figure supplement 1), these "Eomes negative NK" cells referred to as "T. gondii-induced ILC1s" were not directly compared with classical ILC1s. Based on the single cell RNA-seq data, these cells may not express many of the ILC1-related signature genes. Therefore, again, the authors need to be cautious in referring to them as ILC1 cells.

We thank the reviewers for this helpful suggestion. We too had trouble with the best descriptor for these cells and agree that the suggested descriptor is a good one. We have changed the title and the text throughout the paper to now refer to the *T. gondii*-induced cells as “ILC1-like cells,” rather than ILC1s.

A second concern was that the NK 1.1 depletion shown in Figure 1—figure supplement 1 was performed with a PBS rather than isotope matched immunoglobulin control which is considered unacceptable. The authors should repeat at least once with proper control Ig to make sure this is not issue. It is not necessary to repeat entire survival curve just experiments shown in A and B and initial survival to make sure there is no death in controls vs. antibody treated.

We have repeated the experiments shown in Supplementary Figure 1A-B, comparing the effects of NK1.1-depletion with an isotype control antibody (mouse IgG2a isotype control, clone MAR 18.5). (Note that the original description of PK136 as an IgG2b is incorrect. We and others found that it is an IgG2a, and thus is the right control.) The Materials and methods section and figure (Figure 1—figure supplement 1A) have been updated, as well as the description of this experiment (Introduction).

1) Figure 3E-H is focused on the liver, where populations of memory and tissue-resident NK cells reside. Do the authors have data sets for the liver in Figure 3A-C? Do they express Ly6C? Including them in the bar graph of Figure 3C would be useful to identify if the liver trends with the lungs or with the brain (both sites of parasite infection/replication)… or alternatively authors should continue to focus on the lung throughout the entirety of Figure 3.

Original Figure 3 was a large figure with data from different organs that apparently was difficult to follow. For example, the original panels for Figure 3A-C showed markers on cells from lung and blood, and not liver whereas the remaining panels showed other organs. By taking the suggestion of the reviewer, we have thus divided original Figure 3 into two separate figures. New Figure 3 now discusses the lung and blood, while new Figure 3—figure supplement 1 shows other organs. New Figure 4 now focuses on the liver specifically. All other figures have been renumbered to account for this additional figure. Where cited in the rebuttal below, we refer to the new figure numbers, unless otherwise noted.

2) The conclusion of Figure 3F is that "Eomes expression generally remained the same". However, for the representative flow plot of liver NK cells, the authors chose the lowest% of Eomes negative, while in the bar graph the rest of the points show that 20-50% of Eomes^+^ NK cells became Eomes-ve after transfer and migration to the liver. Please reword statement appropriately as Eomes expression is really not the same when the data are pooled.

We rephrased to instead stress how the Eomes expression is high in most transferred NK cells as compared to the transferred ILC1s which mostly remained Eomes-negative (subsection “Expansion of ILC1-like cells among circulating cells”).

3) The interesting point of Figure 3G is the upregulation of CD49a on Eomes^+^ NK cells post-infection. However, this population is eliminated from the subsequent histograms showing Ly6C expression. It may be of interest to note whether Ly6C expression by Eomes^+^CD49a^+^ cells is more similar to ILC1s or NK cells, or occurs at an intermediate level (which would make sense if that population is being seeded from both NK cells and ILC1s).

We have added Ly6C expression by Eomes^+^ CD49a^+^ cells to Figure 4D, as well as a bar graph comparing expression of Ly6C by Eomes^+^ CD49a^–^, Eomes^+^ CD49a^+^, and Eomes^–^ CD49a^+^ cells (Figure 4G). Indeed, Ly6C is expressed at intermediate levels in the Eomes^+^ CD49a^+^ population, suggesting that the Eomes^+^ CD49a^+^ population may be an intermediate. We have added interpretation of the data to the text and updated the figure legends correspondingly (subsection “Expansion of ILC1-like cells among circulating cells”).

4) There are a lot of data represented in Figure 4C-D, without much rationale or interpretation provided in the text suggesting that items may be more appropriate for a supplemental figure.

We have added additional discussion of the significance of the markers CD27 and CD11b as maturation markers (subsection “ILC1-like cells are distinct from NK cells and ILC1s”) and moved these figures to a supplemental figure (Figure 5—figure supplement 1).

5) The CD45.2/.1 appears to be switched on Figure 6—figure supplement 1A.

This was indeed the case. It has been fixed (Figure 6—figure supplement 1).

6) Although the contour plots don't show outliers, the bottom panel of Figure 3E says 3.66%, but there's nothing in the gate, which is a bit misleading.

There are actually points visible within the gate (Figure 4B).

7) The% of Eomes cKO during infection is missing in Figure 6D (bottom).

The percentage has been added to the figure (Figure 7D – 10.3%).

8) The marker Ly6C figures prominently in most of the paper. What is the significance of its expression and of its induction by T. gondii infection? Please provide background.

Previously characterized instances of Ly6C expression by NK cells has been added to the text as well as mention that Ly6C is not expressed by ILC1s in steady-state, at the first mention of Ly6C in the Results section.

9) A recurrent problem is insufficient description of Figures. In several cases (e.g. Figure 4, Figure 5A), the source of cells used for analysis is not given in the Figure, the legend or the text. It would help to include the source of cells, e.g liver or spleen, in each of these places.

The source of cells has been added to the legend and text (Results section and figure legends).

10) As suggested above it would strengthen the manuscript if the authors should characterize the Eomes^+^CD49a^+^ cells found in the liver after T. gondii infection in a greater detail. Do they also express Ly6C?

See response to comment 3 above.

11) In Figure 3F, ~30% of "ILC1s" re-express Eomes upon transfer; this should be discussed.

We have added discussion of this observation to the text (subsection “Expansion of ILC1-like cells among circulating cells”).

12) Figure 3D, the cell labels appear to be switched. Please check.

The cell labels were indeed switched and are now correct (Figure 4A).

13) IL-7Ris an important marker to distinguish NK cells from ILC1s but the cells in cluster 10 do not express IL-7R as shown in Figure 5F. How about IL-2Rb? If possible the dependence of these NK/ILC1 subsets on IL-7 and/or IL-15 should be either tested or discussed.

We have added discussion of the Il7r expression pattern to the text (subsection “*T. gondii* infection induces heterogeneity of NK cells and ILC1s”). Il2rb was highly expressed by all clusters (for reviewers’ reference see Author response image 1) and did not distinguish between clusters and thus was not discussed (same expression level was also seen by flow cytometry of CD122).

14) Figure 3G is confusing. While there are 3 subsets for the combination of CD49a and Eomes expression (upper flow plots) only 2 are shown on the lower panels for Ly6C expression. What are they? Are the colors supposed to match between the top and low panels? Even so, it is not clearly visible.

The 3 subsets refer to Eomes^+^ CD49a^–^ cells, Eomes^+^ CD49a^+^ cells, and Eomes^–^ CD49a^+^ cells. The original figure showed Ly6C expression by Eomes^+^ CD49a^–^ and Eomes^–^ CD49a^+^ cells only, but has been revised to show Ly6C expression of Eomes^+^ CD49a^+^ cells, in response to comments 3 and 10 above (Figure 4D).

15) Subsection “Expansion of ILC1s among circulating cells”. "Ly6C was expressed by a higher percentage." Percentage of what?

We intended to compare Ly6C expression by ILC1-like cells in the spleen versus the liver. This is now clarified in the text (subsection “Expansion of ILC1-like cells among circulating cells”).

16) A number of different markers were used to distinguish among cell subsets, as for example in Figure 4. In many cases, no reason is given for the choice of markers. On what basis were they selected? Besides a phenotypic distinction of subsets, do they contribute to an understanding of the function of these different cells? Please clarify in each case.

Discussion of the significance of the markers KLRG1, DNAM-1, CD11b, and CD27 were added (subsection “ILC1-like cells are distinct from NK cells and ILC1s”).

17) In Figure 4D, lower panels, p values are indicated for different comparisons between markers. The main comparisons should be between uninfected and infected cells, and between NK and ILC1. For what reason are NK cells from uninfected mice compared to ILC from infected mice for every marker measured?

Indeed, we primarily intended to compare the ILC1-like cells from infected mice with the NK cells in infected mice, and the ILC1s in uninfected mice. This has now been corrected (Figure 5, Figure 5—figure supplement 1).

18) Figure 5G, the color gradient, used to presumably show expression level or frequency, does not allow visual discrimination. A different color scale is needed. Some scales go up to 2, others up to 5. What do they indicate?

The Seurat package in R was used for this analysis and this visualization strategy is widely used to show how gene expression is distributed within tSNE plots. Overall, gray indicates lower gene expression and red represents higher levels of gene expression. The limits of the scale varies for each gene shown as the scale is automatically set to maximize contrast between low and high gene expression for each gene, to optimize visual discrimination. The color actually depicts Log2(UMIs+1) and this label has been added next to the scales (Figure 6G).

19) Subsection “NK Cells downregulate Eomes during T. gondii infection”, "the numbers of CD49a^+^ and CD49a^+^ Ly6C^+^ cells were significantly reduced in Eomes cKO compared to Eomes f/f controls (Figure 6A,B)." That result is in 6B. Figure 6A is very different, it shows that the frequency of these cells is higher in Eomes cKO. This difference is not explained. In contrast, the experiment with bone marrow chimeras shown in Figure 6D is very clear. Why are changes in the relative cell frequencies so different from changes in cell numbers?

In the figure (Figure 7A), the cells are pregated on CD3^–^ CD19^–^ NK1.1^+^ NKp46^+^ cells. As conventional NK cells are absent in Eomes cKO mice, the frequency of NK1.1^+^ NKp46^+^ cells that are Ly6C^+^ CD49a^+^ is higher, although the cell number does not change. This has been further clarified in the text (subsection “NK cells downregulate Eomes during *T. gondii* infection”).

20) The columns in Figure 7D are not defined. The corresponding legend is dense. It looks like columns may have been grouped as NK-uninf, NK-inf and ILC1. Does each column represent a different mouse?

The columns are labeled above by color, with each column representing a separate mouse (biological replicates). The labels correspond to the legend shown in Figure 8A, as does the entirety of what is now Figure 8. This was clarified in the figure legend.

21) Subsection “Eomes downregulation within NK cells accompanies extensive transcriptional and epigenomic changes”. "NK cells share many epigenetic features…" Share with whom?

We intended to state that NK cells from uninfected mice and NK cells from infected mice share many epigenetic features. This has been clarified (subsection “Eomes downregulation within NK cells accompanies extensive transcriptional and epigenomic changes.”).

22) Subsection “Eomes downregulation within NK cells accompanies extensive transcriptional and epigenomic changes”. "Somewhat surprisingly" Since T. gondii infection has a large impact on both NK cells and ILC1 cells, why are epigenetic changes common to both populations surprising?

We have deleted “somewhat surprisingly” (subsection “Eomes downregulation within NK cells accompanies extensive transcriptional and epigenomic changes.”).

23) Sequencing analysis shows there are two genes for Ly6C. Does the analysis of phenotypes with anti-Ly6C antibody detect both? What is the significance of having two genes?

There are two distinct genes that encode Ly6C (Ly6c1 and Ly6c2). Currently there are no antibodies that are able to distinguish between the two, and there are no known differences in regulation or expression pattern of the two genes. The significance of there being two discrete genes is not known.

24) Subsection “T. gondii infection results in ILC1 expansion.”. A symbol that is neither an "a" nor a Greek letter α appears in "a-NK1.1". What is it? The context suggests it is an antibody that causes depletion of NK1.1+ cells in mice. If so, it should be described. What does the symbol represent? The symbol appeared again on subsection “T. gondii-induced ILC1s are distinct from NK Cells and ILC1s” in "TNFa". It should be replaced by the Greek letter α.

The symbol in “a-NK1.1” was supposed to refer to “anti,” and has been replaced with “anti.” The alpha symbol (α) is now used in TNFα (Introduction, Results section, Discussion section, Materials and methods section and figure legends).

25) Visually, Figure 8—figure supplement 1C does not convey a clear message. How were the subsets defined? What could be their biological relevance?

The groups are explained in the text in subsection “Eomes downregulation within NK cells accompanies extensive transcriptional and epigenomic changes”. To clarify their significance, their definitions were reiterated in the figure legend and representative schematics of each group were added as Figure 8—figure supplement 1D, with corresponding changes to the legend.

26) The two comparisons made from the ATAC-seq analysis in Figure 7G for Eomes and Figure 8—figure supplement 1F for T-bet show small differences. Is there a way to express a statistical significance? The changes do not look convincing.

Indeed, there was one peak in the Tbx21 locus that was incorrectly highlighted. This highlighted peak has been unhighlighted (Figure 8—figure supplement 1F). Otherwise, significantly different peaks were identified using the getDiffExpression program in Homer, using a Log2FC cutoff of 1 and FDR cutoff of 0.05. These details were added to the Materials and methods section. For the reviewers’ reference, zoomed in images showing all biological replicates with the adjusted p-values for comparison of NK cells from infected mice versus ILC1-like cells at the Eomes and Tbx21 loci are provided in Author response image 2.

**Author response image 2. respfig2:** 

27) The legend to Figure 8—figure supplement 1E does not adequately explain the Figure. What does the figure show?

This figure (now Figure 8—figure supplement 1F) incorporates RNA-seq data with ATAC-seq data to correlate gene expression levels to accessibility. For each locus, each differentially accessible peak is symbolized by a point. The position of the bottom point along the y-axis indicates Log2FC of gene expression by RNA-seq, comparing ILC1-like cells with NK cells from infected mice. The shading of each point represents the Log2FC of each peak’s accessibility, again comparing ILC1-like cells with NK cells from infected mice. Details were added to the figure legend to further clarify. This figure is based off of what was shown in “Epigenetic control of innate and adaptive immune memory” by Lau et al., 2018 in their Figure 6B.